# TASKBENCH: BENCHMARKING LARGE LANGUAGE MODELS FOR TASK AUTOMATION

## ABSTRACT

Recently, the incredible progress of large language models (LLMs) has ignited the spark of task automation, which decomposes the complex tasks described by user instructions into sub-tasks, and invokes external tools to execute them, and plays a central role in autonomous agents. Therefore, there has been an urgent demand to formulate a systematic and standardized benchmark to foster the development of LLMs in task automation. To this end, we introduce TASKBENCH to evaluate task automation. Specifically, the process of task automation can be formulated as three critical stages (i.e., task decomposition, tool invocation, and parameter prediction) to fulfill user intent, that renders its data collection more challenging than common NLP tasks. Here, we introduce the concept of Tool Graph to represent the decomposed tasks in user intent, and adopt a back-instruct method to generate user instruction. Moreover, the mechanism of task automation also drives us to formulate more advanced metrics to measure the capability of LLMs. Therefore, we further propose TASKEVAL to evaluate the capability of LLMs in our curated datasets from different aspects, including task decomposition, tool invocation, and parameter prediction. Experimental results demonstrate that TASKBENCH can effectively be utilized to reflect the capability of LLMs in task automation. The code and datasets of TASKBENCH are available in the supplementary material.

## 1 INTRODUCTION

Due to the recent advances of large language models (LLMs) (Brown et al., 2020; Ouyang et al., 2022; OpenAI, 2023; Touvron et al., 2023a; Anil et al., 2023), LLM-empowered autonomous agents (e.g., AutoGPT (Gravitas, 2023), HuggingGPT (Shen et al., 2023), BabyAGI (Nakajima, 2023), TaskMatrix.AI (Liang et al., 2023)) have unveiled remarkable potential towards artificial general intelligence and become a new rising trend in the realm of AI research. Generally, within the realm of LLM-empowered autonomous agents, task automation is considered as the most important component, which aims to leverage LLMs to autonomously analyze user instructions and accomplish their objectives. Consequently, many researchers attempt to delve deeper into LLM to enable more intelligent task automation. However, it is worth noting that a critical challenge in advancing this area is the lack of a systematic and standardized benchmark to thoroughly evaluate the capability of LLMs in automating tasks. Therefore, creating such a benchmark to facilitate research in this area has become an urgent need.

Nevertheless, it is non-trivial to build such a benchmark for task automation since its setting is closer to real-world scenarios that render data collection and evaluation more challenging than conventional NLP tasks. Figure 1 illustrates a straightforward example to outline the pipeline of task automation, and we can have these observations:

- In contrast to conventional NLP tasks, the procedure to fulfill task automation usually requires multiple stages (e.g., task decomposition, tool invocation, and parameter prediction of tools). This also indicates that we need to take all of these elements into consideration when building benchmark datasets or evaluation metrics;

- In particular, the anticipation of task automation necessitates a broader consideration of real-world scenarios. As a result, the user instruction could be complex. For example, it could be composed of multiple sub-tasks with complex task dependencies. And sometimes, its task scope could demand

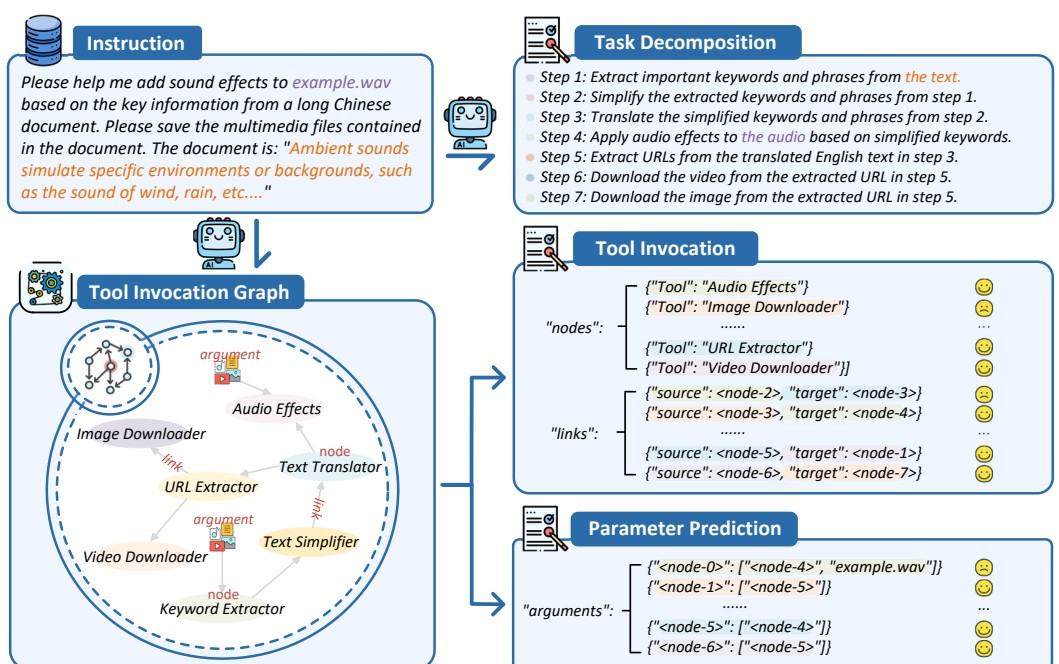

Figure 1: TASKBENCH evaluation for task automation. Task automation implies that LLM-based agents use task decomposition, tool invocation, and parameter prediction to autonomously complete tasks. The evaluation process unfolds as follows: (1) Given a user request, the large language model carries out task decomposition and predicts the tool invocation graph; (2) TaskBench assesses the capability of LLMs in task decomposition based on the decomposed subtasks; 3) For the predicted tool invocation graph, TaskBench evaluates the accuracy of the tool nodes, edges, and parameters.

advanced functions beyond language and thus require the utilization of external tools. These issues also verify the difficulty in constructing the benchmark;

- Furthermore, to better fulfill task automation, it requires LLMs to derive multiple task elements (e.g., decomposed tasks, invoked tools, and tool parameters). These also compel us to devise more advanced evaluation metrics tailored to assess the performance of LLMs in these dimensions.

Therefore, in light of these observations, we found that the majority of existing benchmarks fall short of adequately showcasing the full potential of LLMs in autonomous task completion. For example, conventional benchmarks like GLUE (Wang et al., 2019b) or SuperGLUE (Wang et al., 2019a), encompass a lot of specific tasks to evaluate the capability of LLMs in a single scenario, while cannot well reflect the versatility of task automation. Some other researchers attempted to advocate for more rigorous benchmarks (e.g., MMLU (Hendrycks et al., 2021), GSM8K (Cobbe et al., 2021), AGIEval (Zhong et al., 2023)) by involving more general scenarios (e.g., exams). But all of them can only reflect the capability of language skills, and are not able to manifest the capability of LLMs in task automation. Besides, how to conduct the evaluation for task automation is still a troublesome problem. Therefore, in this paper, we expect to develop a benchmark with appropriate evaluation metrics to better evaluate LLMs in task automation.

To this end, we present TASKBENCH to benchmark the capability of LLMs in the realm of task automation. Specifically, as aforementioned, the data collection for task automation requires us to consider different sophisticated settings, which makes it more challenging. However, just as shown in Figure 1, compared with directly simulating user requests, LLMs usually need to parse tasks for automation, and these parsed tasks (i.e., decomposed tasks, invoked tools, and parameters) are easier to collect and construct. Therefore, we raise a simple idea: *is it possible to synthesize user instruction based on the expected parsed tasks?*

To fulfill this, we first present the concept of Tool Graph (TG), which gathers diverse tools to address specific tasks. Specifically, every two tools in TG can have a connection if they have a dependency. The structure of the whole tool connections can considered as a graph. Therefore, to simulate user

instruction, we can randomly sample a sub-graph from TG to represent the expected task list in user instruction and then apply a back-instruct strategy to generate the final user instruction. During the generation, we provide three different architectures for sampling to enable better controllability, which are node, chain, and directed acyclic graph (DAG). Moreover, a self-critic mechanism is appended to further refine the data quality of our dataset by reviewing their consistency. To guarantee diversity, our data generation is applied to three domains (e.g., Hugging Face (Wolf et al., 2019), multimedia, and daily life) to formulate our TASKBENCH for evaluating LLM in task automation.

After building the dataset, another challenge is how to effectively and quantitatively evaluate the capability of LLMs in task automation. We note that the primary steps of LLMs in automating tasks include task decomposition, tool invocation, and parameter prediction. Therefore, we further propose an evaluation system, called TASKEVAL, which encompasses a series of metrics to provide objective evaluations to measure the capability of LLMs in task decomposition, tool invocation, and predicting the parameters of tools. Moreover, we also conduct human evaluation to prove the positive correlation of our evaluation with human assessment.

Overall, the contributions of our paper can be summarized as:

- We introduce TaskBench, a new benchmark to support the development of LLM in task automation, which comprises a novel data generation to address the data deficiency in this area;
- We further present TASKEVAL to effectively and quantitatively evaluate the capability of LLMs in automating tasks from different aspects, including task decomposition, tool invocation, and parameter predictions;
- The experimental results on different LLMs and additional dataset analysis demonstrate that our proposed TASKBENCH can effectively reflect the capability of LLMs in multiple dimensions with the support of TASKEVAL and show high correlations with human evaluation.

## 2 TASKBENCH

In this section, we introduce the construction of TASKBENCH, the benchmark meticulously designed to facilitate the development of LLMs in task automation. Specifically, unlike previous methods which use collection or instruction methods, TASKBENCH can consider the complex relationships among multiple tasks to simulate more practical and complex user instruction. Figure 2 illustrates the entire process of our method to build the datasets. More details will be introduced in the following subsections.

### 2.1 PRELIMINARY

Task automation aims to fulfill complex user instructions in real-world scenarios. In this setting, the user instructions could encompass multiple sub-tasks, and the execution of each sub-task can be completed by invoking a tool (Schick et al., 2023). Besides, there could also remain some temporal or resource dependencies among these sub-tasks. Therefore, we think that each user instruction can be represented as a combination of tools with connections like graph structure, just as shown in Figure 1. Consequently, we introduce the concept of the Tool Graph (TG), which will be used in our benchmark construction. The tool graph can be viewed as a structured representation that centers on tools with their dependency. Here, we assume a tool as $t$ and denote a TG as $\mathcal{G} = \{T, D\}$, where $T = \{t_1, t_2, \ldots, t_n\}$ represents the collection of tools, and $D$ is a collection of $\{(t_a, t_b)\}$ that means tool $t_a$ exhibits a dependency on tool $t_b$. To some extent, the tool graph offers a novel approach to organizing tools, capturing the relationships between different tools more effectively than traditional taxonomy trees. In the next subsection, we will introduce how to build a tool graph and utilize it to formulate our benchmark.

### 2.2 DATASET CONSTRUCTION

To accomplish user intent, LLMs usually adopt a stepwise process (e.g., task decomposition→tool invocation→parameter prediction) to analyze the user request and convert it into multiple executable tasks. Therefore, it is essential to construct the dataset and allow LLMs to evaluate their automation capability in the above process.

To guarantee that the generated user instructions could cover the expected tasks and dependencies, we adopt a back-instruct strategy to simulate data. More specifically, it can summarized as three steps: 1) we first collect a tool repository and build a tool graph $\mathcal{G}$ with a collection of tools and their dependencies; 2) then we sample a sub-graph from $\mathcal{G}$, to obtain a specified structure; 3) based on the sampled tool sub-graph, we use LLMs to generate user instruction via back-instruct. More details are introduced as below.

### 2.2.1 TOOL GRAPH CONSTRUCTION

Building a tool graph requires us to collect many standalone tools from different sources. When combining different tools together, the dependencies among tools could be diverse, encompassing resource dependencies, temporal dependencies, environment dependencies, and so on. In our research, we mainly investigate two of them: resource and temporal dependencies. For the former one, it means the two tools can have a connection if the input type of tool $t_a$ can match the output type of tool $t_b$. For the latter one, we devise tool graphs that highlight temporal dependencies, allowing any two tools to be linked to illustrate their order. In this work, we choose three scenarios to build the datasets for our benchmark:

**Hugging Face** Hugging Face Wolf et al. (2019) provides a wide variety of AI models to cover massive tasks across language, vision, audio, video, and so on. Each task defined by Hugging Face can be viewed as a tool to address a specific task. Specifically, each tool in Hugging Face has determined the type of its input and output. Hence, if tool $t_a$ and $t_b$ have a connection, the input type of $t_a$ should match the output type of $t_b$. Guided by this principle, we constructed Hugging Face's tool graph, comprising 23 tools and 225 edges.

**Multimedia** In contrast to the Hugging Face tools, which are tailored for AI tasks, the multimedia tools is broader in scope. It provides more user-centric tools like file downloader, video editor, and so on. The policy for tool connections is the same as the Hugging Face domain. Finally, we could construct a tool graph over multimedia tools with 40 nodes and 449 edges.

**Daily Life APIs** Sometimes, we also need some daily life services, including web search, shopping, and etc. Hence, these daily life APIs can also be considered as tools for specific tasks. However, it is worth noting that the type of dependencies among these APIs is predominantly temporal. Therefore, two daily life APIs have a successive order if they are connected. So, in this scenario, we can build a tool graph with 40 nodes and 1,560 edges.

Please note that our method's applicability extends beyond the scenarios mentioned above. We present more details about these tool graphs on different domains in Appendix A.10.

### 2.2.2 SAMPLING ON TOOL GRAPH

Based on the above steps, we can sample a sub-graph from the constructed TG and keep the connections of sampled tools from the TG to capture the dependencies between tools. Following the setting of HuggingGPT, we categorize the sub-structure of a TG into three types: Node, Chain, and directed acyclic graph (DAG). Each type embodies a specific pattern for tool invocation:

- **Node** represents standalone tool invocations, suitable for addressing simple tasks necessitating only a single tool.
- **Chain** corresponds to sequential tool invocations, that tools need to be stepwise executed to complete a task.
- **DAG** depicts more intricate tool invocations. A tool might rely on multiple preceding tools or influence several subsequent tools.

By sampling sub-graphs from these three substructures, we can emulate a variety of valid tool invocation patterns for user instruction. We represent the tool subgraphs in $\mathcal{G}$ as $\mathcal{G}_s = \{T_s, D_s\}$, where $T_s = \{t_{s1}, t_{s2}, \ldots, t_{sk}\}$ with $k < n$ and $D_s = \{(t_{sa}, t_{sb})\}$, such that $t_{sa}$ and $t_{sb}$ belong to $T_s$. The sampling of the tool graph can be described as:

$$\text{Sample}(\mathcal{G}, \text{mode}, \text{size}) \rightarrow \mathcal{G}_s, \tag{1}$$

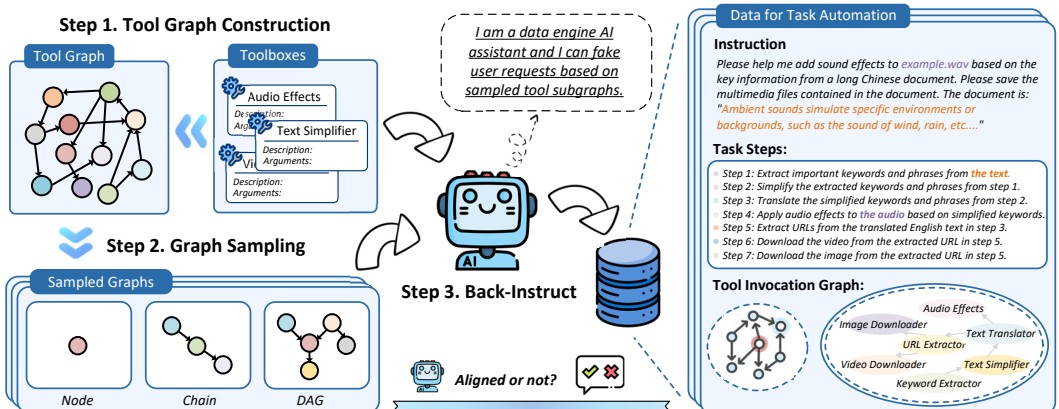

Figure 2: Construction of the TASKBENCH: Initially, we transform the toolbox into a tool graph by creating connections between tools based on their dependencies (either resource or temporal dependencies). Subsequently, we sample diverse subgraphs from the tool graph, which may be single nodes, chains, or directed acyclic graphs. Utilizing the sampled tool subgraphs (which encompass the tools and their interrelations), we "back-instruct" the large language model to inversely craft user instructions, task steps, and tool invocation graphs. Furthermore, we implement critics to evaluate the consistency of the generated tool invocation graphs with the sampled tool subgraphs.

where the mode specifies the sampling mode (e.g., Nodes, Chains, DAGs), and the size indicates the number of tools (Here we set its range as $\{1, 2, ..., 10\}$). These factors determine the topological nature and magnitude of the tool sub-graph in user instructions, respectively.

### 2.2.3 DATA ENGINE WITH BACK-INSTRUCT

Next, based on the sampled sub-graph $\mathcal{G}_s$, we use LLMs to synthesize user instructions. We term this process BACK-INSTRUCT, which can considered as a data engine to convert the sampled tools into user instruction. Specifically, given a sampled subgraph $\mathcal{G}_s$, we formulate the following BACK-INSTRUCT procedure, empowering LLMs to produce the corresponding instructions:

$$\text{BackInstruct}_1(\mathcal{G}_s = (T_s, D_s)) \rightarrow \text{Instruction}. \tag{2}$$

Here, the sampled sub-graph $\mathcal{G}_s$ can instruct LLMs to generate user requests covering these related sub-tasks, and further with their dependencies. Such a strategy ensures the complexity and quality of the generated data.

Specifically, we note that sampled sub-graphs can only provide information on tool invocation skeletons, lacking the critical parameters for tool execution. Therefore, based on the generated instruction in Eqn. 2, we encourage the LLM to populate the parameters for the tool subgraphs and generate the final tool invocation graph along with the corresponding task decomposition steps:

$$\text{BackInstruct}_2(\mathcal{G}_s = (T_s, D_s), \text{Instruction}) \rightarrow \{ \text{Task Steps, Tool Invocation Graph}\}. \tag{3}$$

After that, we introduce a self-critic mechanism to check and filter out the generated instruction to guarantee quality. Here, we offer two variants: LLM-based and rule-based. The former aims to use LLM to check the alignments between the generated data and the sampled tool sub-graph. While the latter uses straightforward rules to determine the alignment between the tool graphs in created data and the sampled tool graphs. Here, we use the nodes and edges of the sampled graph to determine the consistency. Figure 2 illustrates each step of our data engine to simulate user instructions. More detailed designs about our data engine are provided in the Appendix A.8.

### 2.3 DATASET ANALYSIS

Based on the above steps, we build TASKBENCH across three domains, which use GPT-4 as the data engine. The ratio of different modes (i.e., Node, Chain, DAG) is set as $3 : 7 : 8$ for sampling and the ratio for the number of different tools is set as $\{0.1, 0.2, 0.3, 0.2, 0.1, 0.05, 0.025, 0.025, 0.025\}$. Table 1 reports the statistical information of the tool graph and the datasets across three domains.

Notably, it is evident that the two critics we introduced play a crucial role in improving data quality. The rule-based and LLM-based critics respectively filter out an average of 15.13% and 22.73% of the samples. And finally, we obtained 69.22%, 70.54%, and 75.97% of the aligned samples for the three datasets, respectively.

Table 1: Statistics for the TASKBENCH. We report the number of nodes and links of the tool graphs. "# Avg. Nodes" and "# Avg. Links" stands for the average number of nodes and links involved in one sample. We also report the sample number and average request length for the datasets.

| Statistic | Hugging Face Tools | Multimedia Tools | Daily Life APIs |
|---|---|---|---|
| # Nodes of Tool Graph | 23 | 40 | 40 |
| # Links of Tool Graph | 225 | 449 | 1,560 |
| # Avg. Nodes | 3.47 | 3.68 | 3.82 |
| # Avg. Links | 2.46 | 2.68 | 2.8 |
| # Samples | 12,217 | 8,904 | 7,150 |
| - Node / Chain / DAG | 3,270 / 4,302 / 4,645 | 2,117 / 3,145 / 3,642 | 1,277 / 2,716 / 3,157 |
| Avg. Request Length | 41.21 | 39.15 | 38.64 |
| - Node / Chain / DAG | 28.42 / 45.72 / 46.04 | 24.71 / 43.55 / 43.73 | 12.36 / 44.49 / 44.23 |
| self-critic Both critics | 8,456 (69.22%) | 6,281 (70.54%) | 5,432 (75.97%) |
| self-critic LLM-based critic | 9,042 (74.01%) | 6,959 (78.16%) | 5,694 (79.63%) |
| self-critic Rule-based critic | 10,289 (84.22%) | 7,363 (82.69%) | 6,271 (87.70%) |

## 3 TASKEVAL

We could collect ample samples (i.e., synthetic user instructions) with annotations (i.e., sampled tool sub-graph) to evaluate the capability of LLMs in automating tasks. Here, we introduce TASKEVAL, which encompasses a series of evaluation metrics to measure LLMs in multiple dimensions, including task decomposition, tool invocation, and parameter prediction. To simulate the process of LLMs in automating tasks, we adopt a standard prompt for each LLM, which enables it to first disassemble user requests into multiple sub-tasks (i.e., task decomposition), and then predict tool invocations with their parameters and task dependencies to generate a tool invocation graph. Based on the built datasets and the standard inference process, we design pertinent metrics to evaluate three stages (i.e., task decomposition, tool invocation, and parameter predictions) in task automation. Here, we choose the GPT family (Brown et al., 2020; Ouyang et al., 2022; OpenAI, 2023) and open-source LLMs (Touvron et al., 2023a; Chiang et al., 2023; Rozière et al., 2023; Xu et al., 2023; Yang et al., 2023) as our main evaluation. Please see the Appendix A.7 for full evaluations of other open-source LLMs (Team, 2023a; Li et al., 2023a; Team, 2023b).

### 3.1 TASK DECOMPOSITION

Task decomposition is a pivotal component of task automation. By decomposing user instruction into a sequence of executable sub-tasks, the autonomous agent can more effectively fulfill user intent. During the task decomposition, each step will generate textual descriptions. Here, we use three subjective metrics to measure the quality in analyzing sub-tasks: **Rouge-1 *(R1)***, **Rouge-2 *(R2)***, and **BertScore F1 *(BsF)*** (Zhang et al., 2019). The results are reported in Table 2. We observe that the GPT-4 model significantly outperforms the open-source LLM model in task decomposition, achieving approximately 20%+ higher than others in Rouge-1/2. Moreover, we find that codellama-13b achieves the closest performance to the GPT family. We attribute the substantial code pre-training endowing it with a higher-level task decomposition capability compared to other LLMs.

### 3.2 TOOL INVOCATION

The graph of tool invocation can be viewed as a concrete representation of task steps in user instruction, specifying the appropriate tool for each step. To orchestrate external tools effectively, the tool invocation graph should provide these pieces of information: 1) the dependency between tools to guarantee the order of executable sub-tasks; 2) the parameters that tools require. Therefore, it

Table 2: Evaluation for task decomposition. We compare the text descriptions between the generated and real task steps in terms of Rouge-1 (R1), Rouge-2 (R2), and BertScore F1 (BsF).

| | TASK DECOMPOSITION - Step-by-step task decomposition | | | | | | | | |
|---|---|---|---|---|---|---|---|---|---|
| **LLM** | **Hugging Face Tools** | | | **Multimedia Tools** | | | **Daily Life APIs** | | |
| | *R1* ↑ | *R2* ↑ | *BsF* ↑ | *R1* ↑ | *R2* ↑ | *BsF* ↑ | *R1* ↑ | *R2* ↑ | *BsF* ↑ |
| gpt-4 | 52.56 | 30.49 | 90.12 | 60.92 | 40.02 | 91.17 | 82.93 | 68.94 | 96.49 |
| gpt-3.5-turbo | 43.31 | 21.83 | 88.48 | 50.08 | 28.74 | 89.54 | 54.82 | 34.87 | 90.26 |
| text-davinci-003 | 37.04 | 17.87 | 87.04 | 49.53 | 28.08 | 89.20 | 62.62 | 42.10 | 91.80 |
| codellama-13b | 39.08 | 18.62 | 88.32 | 44.95 | 23.57 | 88.68 | 67.79 | 48.68 | 92.97 |
| vicuna-13b-v1.5 | 37.48 | 17.25 | 87.90 | 45.07 | 23.89 | 88.91 | 54.50 | 34.23 | 90.29 |
| nous-hermes-13b | 37.71 | 17.15 | 88.18 | 36.26 | 16.40 | 87.53 | 49.23 | 30.13 | 89.41 |
| baichuan-13b-chat | 19.99 | 6.01 | 83.83 | 20.37 | 3.71 | 83.21 | 50.63 | 27.85 | 88.39 |
| wizardlm-13b | 34.77 | 15.58 | 87.37 | 36.34 | 17.80 | 87.29 | 47.48 | 24.06 | 88.22 |
| llama-2-13b-chat | 39.77 | 18.93 | 88.69 | 26.09 | 7.76 | 84.73 | 46.78 | 23.23 | 87.82 |

Table 3: Evaluation for parameter prediction of tools. *t-F1* evaluate the pair of (task, parameter name), *v-F1* evaluate the triple of (task, parameter name, parameter value).

| | | TOOL PARAMETER PREDICTION - Predicts parameters for the tool execution. | | | | | | | |
|---|---|---|---|---|---|---|---|---|---|
| | **LLM** | **Node** | | **Chain** | | **DAG** | | **Overall** | |
| | | *t-F1* ↑ | *v-F1* ↑ | *t-F1* ↑ | *v-F1* ↑ | *t-F1* ↑ | *v-F1* ↑ | *t-F1* ↑ | *v-F1* ↑ |
| **Hugging Face Tools** | gpt-4 | 79.46 | 74.03 | 76.26 | 56.89 | 77.47 | 58.39 | 77.10 | 59.06 |
| | gpt-3.5-turbo | 37.26 | 19.69 | 60.68 | 39.92 | 61.78 | 41.70 | 58.30 | 38.05 |
| | text-davinci-003 | 38.24 | 27.54 | 57.21 | 37.84 | 58.36 | 39.07 | 55.16 | 36.99 |
| | nous-hermes-13b | 46.06 | 30.99 | 34.78 | 12.70 | 35.25 | 12.43 | 36.25 | 14.89 |
| | wizardlm-13b | 43.66 | 26.20 | 36.63 | 11.33 | 37.80 | 12.21 | 37.94 | 13.43 |
| | llama-2-13b-chat | 29.49 | 20.65 | 32.38 | 13.35 | 33.01 | 14.56 | 32.29 | 14.89 |
| | codellama-13b | 19.85 | 12.62 | 36.70 | 20.80 | 36.05 | 20.99 | 33.99 | 19.59 |
| | vicuna-13b-v1.5 | 25.60 | 13.32 | 29.10 | 10.80 | 30.91 | 12.07 | 29.46 | 11.73 |
| | baichuan-13b-chat | 45.95 | 29.69 | 30.02 | 9.12 | 30.48 | 9.72 | 31.83 | 11.63 |
| **Multimedia Tools** | gpt-4 | 94.48 | 86.75 | 85.85 | 70.53 | 85.92 | 69.16 | 86.46 | 70.93 |
| | gpt-3.5-turbo | 44.04 | 11.60 | 71.09 | 49.07 | 71.35 | 47.60 | 68.39 | 44.23 |
| | text-davinci-003 | 59.93 | 20.27 | 70.95 | 47.80 | 70.53 | 45.46 | 69.68 | 43.97 |
| | nous-hermes-13b | 49.99 | 37.94 | 40.93 | 16.40 | 41.21 | 16.04 | 41.94 | 18.38 |
| | wizardlm-13b | 49.97 | 33.95 | 35.52 | 13.36 | 36.73 | 14.32 | 37.51 | 15.97 |
| | llama-2-13b-chat | 28.73 | 17.28 | 30.20 | 9.06 | 31.24 | 9.64 | 30.53 | 10.32 |
| | codellama-13b | 31.68 | 15.87 | 51.00 | 31.39 | 50.73 | 30.13 | 48.60 | 28.89 |
| | vicuna-13b-v1.5 | 52.08 | 35.33 | 36.81 | 18.93 | 38.11 | 19.68 | 38.86 | 20.94 |
| | baichuan-13b-chat | 40.76 | 28.40 | 24.71 | 7.97 | 25.71 | 8.14 | 26.59 | 9.89 |
| **Daily Life APIs** | gpt-4 | 93.99 | 76.26 | 97.52 | 71.90 | 96.23 | 68.92 | 96.71 | 70.61 |
| | text-davinci-003 | 61.95 | 45.01 | 82.61 | 55.87 | 81.39 | 53.64 | 80.92 | 54.18 |
| | gpt-3.5-turbo | 42.88 | 27.12 | 89.25 | 60.48 | 88.34 | 57.98 | 85.05 | 56.38 |
| | nous-hermes-13b | 30.31 | 23.35 | 41.22 | 27.96 | 39.13 | 26.26 | 39.52 | 26.84 |
| | wizardlm-13b | 44.67 | 35.26 | 54.84 | 36.84 | 54.64 | 35.76 | 53.99 | 36.20 |
| | llama-2-13b-chat | 10.38 | 7.29 | 39.51 | 25.63 | 37.98 | 24.08 | 36.80 | 23.62 |
| | codellama-13b | 50.01 | 38.35 | 65.20 | 45.55 | 63.11 | 42.84 | 63.16 | 43.73 |
| | vicuna-13b-v1.5 | 20.14 | 14.63 | 51.10 | 35.44 | 50.99 | 34.04 | 49.73 | 33.84 |
| | baichuan-13b-chat | 32.86 | 22.19 | 37.91 | 23.72 | 38.03 | 22.15 | 37.65 | 22.83 |

is necessary for us to measure the capability of LLMs in these aspects. Here, we first evaluate the predicted graph structure to measure the capability of LLMs in this subsection.

In a tool invocation graph, we denote the nodes as tools and the edge as the dependency between two tools. Therefore, we can evaluate the nodes and edges to measure the capability of LLMs in tool invocation. Here, we crafted two distinct metrics: **Node F1 *(n-F1)*** for node prediction and

Table 4: Evaluation for tool invocation. Node F1 *(n-F1)* for node prediction and Edge F1 *(e-F1)* for edge prediction. For nodes, a prediction is deemed positive if the predicted node's ID aligns with any of the ground-truth node labels. For edges, both the source and target nodes of a predicted edge must correspond exactly. Normalized Edit Distance *(NED)* measures the normalized number of operations required to correct the prediction for chain structure.

| | | **Node** | **Chain** | | | **DAG** | | **Overall** | |
| | **LLM** | n-F1 ↑ | n-F1 ↑ | e-F1 ↑ | NED ↓ | n-F1 ↑ | e-F1 ↑ | n-F1 ↑ | e-F1 ↑ |
|---|---|---|---|---|---|---|---|---|---|
| **Hugging Face Tools** | gpt-4 | 84.34 | 80.35 | 56.78 | 43.95 | 82.14 | 58.13 | 81.37 | 56.18 |
| | gpt-3.5-turbo | 56.89 | 72.15 | 40.74 | 50.03 | 74.22 | 42.02 | 69.81 | 35.31 |
| | text-davinci-003 | 40.65 | 66.84 | 36.98 | 51.53 | 66.00 | 37.27 | 60.44 | 31.16 |
| | codellama-13b | 43.65 | 55.86 | 18.70 | 64.10 | 54.90 | 16.39 | 53.56 | 15.60 |
| | baichuan-13b-chat | 58.34 | 52.88 | 7.66 | 63.10 | 52.51 | 8.33 | 53.69 | 7.52 |
| | nous-hermes-13b | 58.67 | 52.04 | 8.74 | 64.52 | 53.13 | 7.45 | 53.51 | 8.02 |
| | vicuna-13b-v1.5 | 51.76 | 50.15 | 8.02 | 68.68 | 52.53 | 9.54 | 51.03 | 7.43 |
| | llama-2-13b-chat | 43.58 | 50.14 | 8.39 | 66.44 | 50.05 | 8.30 | 48.82 | 7.49 |
| | wizardlm-13b | 54.67 | 54.46 | 1.98 | 62.55 | 54.90 | 1.84 | 54.60 | 1.91 |
| **Multimedia Tools** | gpt-4 | 97.13 | 88.92 | 71.84 | 35.15 | 90.49 | 72.72 | 90.27 | 71.88 |
| | gpt-3.5-turbo | 53.55 | 76.91 | 52.91 | 43.79 | 76.89 | 51.44 | 73.23 | 47.07 |
| | text-davinci-003 | 59.15 | 77.58 | 54.53 | 43.03 | 75.59 | 50.58 | 74.17 | 48.96 |
| | codellama-13b | 43.70 | 66.44 | 28.84 | 50.60 | 67.36 | 27.94 | 62.98 | 25.12 |
| | vicuna-13b-v1.5 | 66.64 | 58.04 | 15.94 | 57.73 | 60.50 | 16.56 | 59.96 | 14.97 |
| | nous-hermes-13b | 60.58 | 57.99 | 8.46 | 59.02 | 58.52 | 9.16 | 58.53 | 8.25 |
| | baichuan-13b-chat | 45.59 | 41.69 | 4.61 | 66.09 | 41.22 | 6.08 | 42.06 | 4.96 |
| | wizardlm-13b | 55.13 | 50.55 | 4.12 | 61.28 | 50.01 | 5.41 | 51.09 | 4.39 |
| | llama-2-13b-chat | 38.02 | 45.36 | 1.43 | 67.23 | 45.44 | 1.83 | 44.16 | 1.52 |
| **Daily Life APIs** | gpt-4 | 94.39 | 97.63 | 78.40 | 40.12 | 97.08 | 68.84 | 97.23 | 70.56 |
| | gpt-3.5-turbo | 50.74 | 90.69 | 65.95 | 44.66 | 89.69 | 39.18 | 86.45 | 51.77 |
| | text-davinci-003 | 68.49 | 83.68 | 62.62 | 47.79 | 82.20 | 33.89 | 82.13 | 50.75 |
| | codellama-13b | 59.10 | 72.85 | 47.31 | 54.05 | 70.51 | 25.53 | 70.85 | 37.41 |
| | vicuna-13b-v1.5 | 40.32 | 67.04 | 31.36 | 61.16 | 65.34 | 17.30 | 65.32 | 24.88 |
| | wizardlm-13b | 61.33 | 62.57 | 25.05 | 58.09 | 62.07 | 14.73 | 62.30 | 19.96 |
| | nous-hermes-13b | 37.02 | 53.43 | 20.17 | 65.85 | 51.14 | 10.61 | 51.42 | 15.78 |
| | llama-2-13b-chat | 34.11 | 58.21 | 20.56 | 67.24 | 56.35 | 11.70 | 56.26 | 15.73 |
| | baichuan-13b-chat | 52.50 | 51.56 | 11.05 | 70.36 | 53.74 | 7.99 | 52.41 | 9.51 |

**Edge F1** *(e-F1)* for edge prediction. We also introduced the **Normalized Edit Distance** *(NED)* metric for chain structure, quantifying the adjustments needed to correct predictions. The results are detailed in Table 4 revealing that predicting edges in the tool invocation graph is more challenging than predicting nodes, with an F1 score difference of about 30% across all LLMs. Furthermore, we also observe that tasks with varying structures pose different challenges for LLMs. For instance, on simple node structures, open-source LLMs match the performance of gpt-3.5-turbo and text-davinci-003, but lag behind on more complex tasks. Overall, these designed metrics can effectively help us to better measure the capability of LLMs in task automation.

## 3.3 PARAMETER PREDICTION

We further divide the evaluation of the tool parameter prediction as twofold. The first metric is **Parameter Name F1** *(t-F1)* to evaluate the capability of LLMs in predicting the parameter of the tools. Another one is the **Parameter Name & Value F1** *(v-F1)* to measure both the parameter and its value. The results can be found in Table 3. We found that the precise prediction of parameters can determine the successful execution of tools, and the precise prediction of both parameters and values can guarantee the correctness of the tool invocation. Besides, we found that GPT-4 can significantly outperform open-source LLMs, which achieve *v-F1* scores of 59.06%, 70.93%, and 70.61% across the three domains. These results also highlight the limitations of current open-source LLMs in task automation, and which parts should be enhanced.

Table 5: Alignment of TASKBENCH with human evaluation. Kendall's $\tau$ alculates the proportion of aligned pairs, while Spearman's $\rho$ measures the correlation between the ranks of elements.

| Correlation Metric | Hugging Face Tools | Multimedia Tools | Daily Life APIs | Average |
|---|---|---|---|---|
| Kendall's $\tau$ | 0.89 | 0.83 | 0.94 | 0.89 |
| Spearman's $\rho$ | 0.78 | 0.62 | 0.93 | 0.78 |

## 3.4 HUMAN EVALUATION

To ensure the reliability of TASKBENCH evaluations, we further investigate their consistency with human evaluations. We randomly select 50 samples from the constructed instructions and then enforce the evaluated LLMs to generate responses. We then compares pairwise LLMs and summarizes their ranking based on the executability and problem-solving efficacy of the tool invocation graphs. To illustrate the alignment between TASKBENCH and human evaluations, we used two metrics: Kendall's $\tau$ and Spearman's $\rho$. The results are shown in Table 5. We find that the average values for Kendall's $\tau$ and Spearman's $\rho$ are 0.89 and 0.78, respectively. This indicates a very positive correlation between human evaluation and our TASKBENCH, which further validates the effectiveness of our proposed framework for dataset construction. We also conduct a human evaluation and a case study on the constructed dataset. Please refer to Appendix A.3 and Appendix A.4 for more details.

## 3.5 ANALYSIS

**Factors Contributing to Task Automation Performance**  Our analysis identifies two primary factors impacting the performance of LLMs in task automation: 1) *Reasoning Capabilities*: LLMs vary significantly in their abilities to solve complex problems and reason effectively, which are crucial for task automation. For instance, GPT series demonstrate superior reasoning abilities in math and coding tasks, which indicates enhanced skills in task planning and tool utilization. 2) *Instruction Following*: Our analysis reveals that models fine-tuned with instruction-following settings, such as Vicuna-13b, WizardLLM-13b, and Nous-Hermes-13b, outperform the baseline Llama-2-13b model in task automation. Furthermore, WizardLLM-13b can perform better than Vicuna-13b due to more complex instruction fine-tuning. These results also demonstrate the necessity of instruction following.

**Intrinsic Differences in LLMs in Performing Task Automation**  Our findings suggest that: 1) *Code Pre-training*: We note that the models (Code-Llama) with more code-pretraining outperform other open-source LLMs in task automation. Experimental results show an average improvement of 4.45% in tool prediction (*n-f1*) and 12.76% in parameter prediction (*v-f1*) across various domain datasets. We conclude that task automation usually involves multiple stages which makes it need structural text as the interface to connect different stages; 2) *Alignment Techniques*: Besides, the models (e.g., GPT series models) with human alignments (e.g., RLHF) demonstrate stronger task automation capabilities than open-source LLMs. These results also indicate that RLHF allows large language models to develop more generalized reasoning abilities, reducing overfitting to specific instructions.

## 4 CONCLUSION

In this paper, we introduce TaskBench, a benchmark to evaluate LLMs for task automation. More in detail, we first summarize three critical stages for LLMs in automating tasks, which include task decomposition, tool invocation, and parameter prediction for tools. The performance of these three stages reflects the task automation capabilities of LLMs, and thus we expect to construct evaluation datasets for them. To fulfill this, we introduce the concept of ToolGraph, which collects different tools with their connections, and then adopt a back-instruct method to simulate user instruction based on the sampled sub-graph from ToolGraph. Based on our curated datasets, we further introduce TASKEVAL to systematically evaluate the capability of LLMs in automating tasks, including task decomposition, tool invocation, and parameter prediction. Experimental results demonstrate our TaskBench can be effectively utilized to evaluation LLMs in task automation. In the future, we will extend our benchmark to various domains and design more advanced metrics to further explore the potential of LLMs in task automation and build powerful autonomous agents.

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

# A APPENDIX

## A.1 RELATED WORKS

Large language models (ChatGPT (Ouyang et al., 2022), GPT-4 (OpenAI, 2023), LLAMA (Touvron et al., 2023a;b), Bard (Anil et al., 2023)) have drawn the development of autonomous agents (e.g., AutoGPT (Gravitas, 2023), HuggingGPT (Shen et al., 2023), BabyAGI (Nakajima, 2023)). These applications can be considered as a form of task automation, which uses LLMs as the controller to analyze user instructions and search for the most suitable solution (e.g., external models) to obtain answers. Despite these advances in this area, it still lacks a systematic and standardized benchmark to measure the capability of LLMs in automation tasks. Traditional benchmarks like GLUE (Wang et al., 2019b) or SuperGLUE (Wang et al., 2019a) can only evaluate the capability of pre-trained models in a single task. To further explore the capability of LLMs, some benchmarks (AlpacaEval (Li et al., 2023b), AGIEval (Zhong et al., 2023), GSM8K (Cobbe et al., 2021), MMLU (Hendrycks et al., 2021) and etc) are introduced to explore the capability of LLMs in solving problems, but they can only manifest the language skills and are inadequate to test the capability in automating tasks. Recently, some works attempted to introduce some benchmarks to facilitate the development of this area. For example, APIBench is proposed by Gorilla (Patil et al., 2023) which uses self-instruct (Wang et al., 2023) to generate data from API calls. ToolAlpaca (Tang et al., 2023), ToolQA (Zhuang et al., 2023) and ToolBench (Qin et al., 2023) respectively introduce different benchmarks from the perspective of tool utilization. Besides, AgentBench (Liu et al., 2023) is a benchmark to explore the capability of LLMs in different environments, which is another important part of autonomous agents. In this paper, we focus on multiple dimensions (e.g., task decomposition, tool invocation, and parameter prediction) to build a multifaceted benchmark, including dataset and evaluation, to comprehensively evaluate the capability of LLMs in task automation.

## A.2 DISCUSSION WITH RELATED WORKS

We also note that there are some concurrent works (e.g., ToolBench, APIBench, etc) to investigate this area. Therefore, we also conducted an in-depth discussion to analyze the differences between TaskBench and these works. Generally, we summarize these differences from the perspectives of datasets and evaluations:

**Datasets** Currently, all of these works will adopt a self-instruct strategy to generate instructions to simulate complex user instructions, which requires planning, tool utilization, and etc. However, most of them still lack the annotations of ground truth, and only rely on human evaluation to check the quality of LLM generations. Some works like ToolBench additionally use a depth-first search-based decision tree to search a solution path annotation. But such a strategy will also bring some new issues like costs and hallucination or bias in annotations. Compared with them, due to the design of the Tool Graph, TASKBENCH can guarantee the ground truth and demonstrate these advantages:

- **Efficiency**: Our method requires only one API call to generate a complete sample (creating instructions, generating a tool invocation graph, and checking). In contrast, ToolLLM requires one API call to generate instructions and an average of four inference steps during DFS for annotation. Additionally, ToolLLM uses few-shot learning, which consumes more tokens than our zero-shot approach.

- **Higher Quality of Instructions**: Our tool graph includes potential tool invocation relationships (resource and temporal dependencies). Based on the tool graph, the instructions generated by ChatGPT are more in line with actual human behavior. As we demonstrated in Appendix A.3, the generation based on a tool graph with edges significantly improves Naturalness, Complexity, and Alignment compared to generation without edges.

- **Stability**: ToolLLM's instruction generation might not cover all sampled instructions based on the API set. Our method not only covers all sampled tools but also strictly follows the dependency between tools. We can control instruction generation through the sampling process, such as the number of tools and the topology of the tool graph.

- **Reliable Annotations**: In ToolLLM, instruction generation, and tool invocation annotation are independent processes. However, in our approach, the sampled tool graph can directly be utilized as the final annotations for the assessment. Hence, our Back-Instruct can directly generate

instructions with better alignments to the annotations and does not need to generate answers anymore. Besides, we also utilize the consistency between the sampled tool graph and the generated tool invocation graph to further filter out low-quality annotations, ensuring high-quality tool graph annotations.

**Evaluation**  During the evaluation, most of them usually use human evaluation to measure the capability of LLMs, which requires massive human labor. Some works (e.g., TOOLBENCH) follow the AlpacaEval setting to measure the capability of LLMs in tool utilization. Compared with these settings, Our evaluation is composed of both subjective and objective evaluations, which also encompasses multiple aspects, including task decomposition, tool prediction, and parameter prediction. For each aspect, we have developed multiple well-designed metrics to assess the different capabilities required for task automation. Besides, we also conduct human evaluations to investigate the performance of different LLMs and demonstrate the correlations with the subjective evaluation metrics.

### A.3  HUMAN EVALUATION

To demonstrate the quality of TASKBENCH, we conducted in-depth human evaluations based on generated samples.

**Evaluation Metrics**  To assess the quality of datasets constructed by Back-Instruct, we designed three metrics in our evaluation criteria. Two measure the quality of instructions, and one evaluates tool invocation graphs:

- Metrics for Instruction:
  - **Naturalness:** This metric measures the reasonableness of the instructions, including the commonality of dependencies between tools and their alignment with real-world needs.
  - **Complexity:** This metric assesses the complexity of the instructions, considering factors such as task depth, the number of involved tools, and the relationships between these tools.
- Metric for Tool Invocation Graphs:
  - **Alignment:** Building upon the Feasibility metric, this measures how well the tool invocation graphs align with the instructions, i.e., whether the tool invocation graphs can effectively address the user's commands.

Each metric is scored from 1 to 5, and we design these metrics to assess the effectiveness and faithfulness of our TASKBENCH in task automation.

**Comparison with Baselines**  To make a fair comparison, we choose two additional baselines to compare our Back-Instruct:

- **Back-Instruct (Ours):** we sample tool subgraphs and then backtranslate to instructions and further refine the tool invocation graph.
- **Back-Instruct w/o edges:** compared with our Back-Instruct, we eliminated edges from our sampled tool subgraphs, preserving only the tool node information in the prompt.
- **Self-Instruct:** based on manually labeled demonstrations and all tools with descriptions, we directly employed GPT-4 to autonomously select tools and then generate the instructions with tool invocation graphs.

**Evaluation Results**  During the human evaluation, we randomly selected 50 samples from our TASKBENCH and invited three domain experts to assess the quality of these samples. To ensure a fair and unbiased evaluation, all samples will be anonymized. We provide canonical samples for these experts to calibrate their criteria during the annotations, and calculate an average score of all experts' ratings as the final results. All results can be found in Table 6.

We observed that all methods (Self-Instruct or Back-Instruct) can guarantee the alignment. However, our method, Back-Instruct, scored highest in Naturalness and Complexity. We attribute these superiorities to the realistic resource or temporal dependencies in the sampled tool subgraphs, which allow us to generate more natural instructions in complex scenarios (e.g., multi-tools utilization).

Table 6: Human evaluation (rating from 1 to 5) on samples constructed by different methods. Average score rating from three human experts.

| Methods | Naturalness↑ | Complexity↑ | Alignment↑ | Overall↑ |
|---|---|---|---|---|
| Back-Instruct | 3.89 | 4.01 | 3.66 | 3.85 |
| Back-Instruct w/o edges | 3.44 | 3.27 | 3.62 | 3.44 |
| Self-Instruct | 2.18 | 2.01 | 3.64 | 2.61 |

**Case Analysis**  Moreover, we further draw some cases to intuitively show the differences between the three methods, as shown in Table 7. From these examples, we observe that our Back-Instruct can generate examples with more comprehensive and interconnected tool usage, reflecting higher naturalness and complexity in instruction generation.

## A.4  ERROR ANALYSIS

### A.4.1  ERROR ANALYSIS ON TASKBENCH DATASET

Despite the advanced instruction generation and labeling capabilities of gpt-4, we admit that it is challenging to guarantee the correctness of all generated samples. To better understand our dataset and assess its accuracy, we conduct human evaluations to provide a thorough error analysis. Here, we first randomly sampled 148 samples, and our labeling team identified 18 error samples (nearly 12%) from the sampled data. We attribute these incorrect samples to five distinct error categories. Typical examples and the proportions for each category are shown in Table 8 and Figure 3:

- **Incorrect Instructions**:
  - **Incomplete instructions**: This error occurs when the instructions lack the necessary details or resources for successful completion.
  - **Impractical instructions**: The instructions could be irrational or impossible to execute with the capabilities of current tools.
- **Parameter Errors:**
  - **Mismatched parameter types**: This error occurs when the parameters provided do not match the expected types for the used tool.
  - **Incorrect parameter value**: This error is evident when the values provided for the parameters are incorrect or not suitable for the task.
- **Incorrect Tool Dependency**: This error type refers to the incorrect linking or sequencing of tools required for a task.

Based on these observed errors, we conclude that it is necessary to build a more elaborate prompt (e.g., more detailed tool-use specification and demonstrations) to describe tool parameters and tool dependencies when generating the tool invocation graph. Besides, we will also introduce more high-quality criteria to continuously improve our dataset in addition to our rule-based and LLM-based critics.

### A.4.2  ERROR ANALYSIS OF DIFFERENT LLMS IN PREDICTING TOOL INVOCATION GRAPH

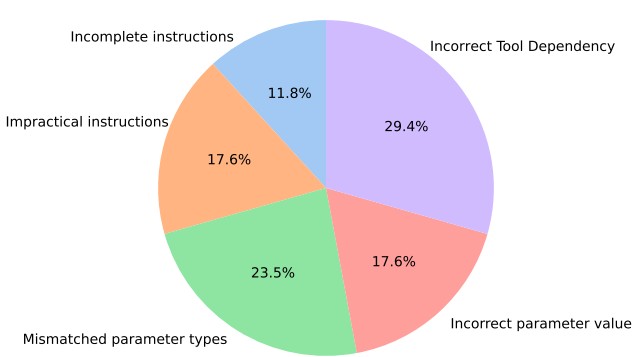

Figure 3: Error Analysis on TASKBENCH

We analyze the failures in predicting the tool invocation graph that occur during task automation inference. These failures can be categorized into three main groups: incorrect tool names, incorrect tool dependencies, and mismatched

Table 7: Comparative analysis of Back-Instruct, Back-Instruct w/o edges, and Self-Instruct.

| Method | Tools | Instruction | Tool Invocation Graph |
|---|---|---|---|
| Back-Instruct | `"sampled_nodes": [{"id": "Audio-to-Image"}, {"id": "Image Colorizer"}, {"id": "Image Stitcher"}, {"id": "Image Style Transfer"}, {"id": "Image-to-Text"}], "sampled_links": [{"source": "Image Style Transfer", "target": "Image Stitcher"}, {"source": "Audio-to-Image", "target": "Image Colorizer"}, {"source": "Image Colorizer", "target": "Image Style Transfer"}]` | I have an image (example.jpg) with some text on it and an audio file (example.wav). I'd like you to extract the text from the image, generate a visual representation of the audio file, colorize this visual representation using the colors from the original image, apply the visual style of the colorized image to the visual representation of the audio, and finally, stitch these two images together to create a panorama or collage. | `{"nodes": [{"id": "Audio-to-Image", "input": ["example.wav"]}, {"id": "Image Colorizer", "input": ["Audio-to-Image"]}, {"id": "Image Stitcher", "input": ["example.jpg", "Image Style Transfer"]}, {"id": "Image Style Transfer", "input": ["Image Colorizer", "Audio-to-Image"]}, {"id": "Image-to-Text", "input": ["example.jpg"]}], "links": [{"source": "Audio-to-Image", "target": "Image Colorizer"}, {"source": "Image Colorizer", "target": "Image Style Transfer"}, {"source": "Image Style Transfer", "target": "Image Stitcher"}]}` |
| Back-Instruct w/o edges | `"sampled_nodes": [{"id": "Audio-to-Image"}, {"id": "Image Colorizer"}, {"id": "Image Stitcher"}, {"id": "Image Style Transfer"}, {"id": "Image-to-Text"}]` | I would like to create a stylized collage from 'example.jpg' and the sound of the 'example.wav' file. The visual style should be inspired by the similar image search result. The collage should be colorized, and I would like any text in the final image to be extracted. | `{"nodes": [{"id": "Audio-to-Image", "input": ["example.wav"]}, {"id": "Image Colorizer", "input": ["Image Style Transfer"]}, {"id": "Image Search (by Image)", "input": ["example.jpg"]}, {"id": "Image Stitcher", "input": ["Audio-to-Image", "example.jpg"]}, {"id": "Image Style Transfer", "input": ["Image Stitcher", "Image Search (by Image)"]}, {"id": "Image-to-Text", "input": ["Image Colorizer"]}], "links": [{"source": "Audio-to-Image", "target": "Image Stitcher"}, {"source": "Image Colorizer", "target": "Image-to-Text"}, {"source": "Image Search (by Image)", "target": "Image Style Transfer"}, {"source": "Image Stitcher", "target": "Image Style Transfer"}, {"source": "Image Style Transfer", "target": "Image Colorizer"}]}` |
| Self-Instruct | `"sampled_nodes": [{"id": "Audio-to-Image"}, {"id": "Image Colorizer"}, {"id": "Image Stitcher"}, {"id": "Image Style Transfer"}, {"id": "Image-to-Text"}]` | I have a black and white image of an old newspaper (example.jpg) and I want to colorize it first, then apply the style of a modern newspaper (example.png) to it, and finally extract the text from the stylized image. | `{"nodes": [{"id": "Image Colorizer", "input": ["example.jpg"]}, {"id": "Image Style Transfer", "input": ["Image Colorizer", "example.png"]}, {"id": "Image-to-Text", "input": ["Image Style Transfer"]}], "links": [{"source": "Image Colorizer", "target": "Image Style Transfer"}, {"source": "Image Style Transfer", "target": "Image-to-Text"}]}` |

tool parameters. For our analysis, we randomly selected 50 predictions, and the distribution of each error type across different LLMs is detailed in Table 9. We observed that:

• gpt-4 demonstrates the fewest errors in all categories, indicating a higher accuracy in predicting the tool invocation graph.

Table 8: Error Analysis on TASKBENCH.

| Error Type | Example |
|---|---|
| Incomplete instructions | I have a long text and I would like to get a summarized version of it, then generate an image that represents the main idea of the summarized text. |
| Impractical instructions | I have a text: 'This training vid is amazing! Speed it up by 1.5x please!'. Analyze the sentiment, expand it, find the video URL and adjust the video speed. |
| Mismatched parameter types | I want to find articles related to climate change and analyze their sentiment. Please translate non-English articles to English. `{"task_nodes": [{"task": "Text Search", "arguments": ["climate change"]}, {"task": "Text Sentiment Analysis", "arguments": ["<node-0>"]}, {"task": "Text Translator", "arguments": ["<node-1>"]}], "task_links": [{"source": "Text Search", "target": "Text Sentiment Analysis"}, {"source": "Text Sentiment Analysis", "target": "Text Translator"}]}` |
| Incorrect parameter value | I have two audio files from online lectures at the following URLs: 'example1.wav' and 'example2.wav'. I want them combined into a single audio file, transcribe the speech into text, and check the text for grammatical errors. `{"task_nodes": [{"task": "Audio Downloader", "arguments": ["example1.wav", "example2.wav"]}, {"task": "Audio Splicer", "arguments": ["<node-0>"]}, {"task": "Audio-to-Text", "arguments": ["<node-1>"]}, {"task": "Text Grammar Checker", "arguments": ["<node-2>"]}], "task_links": [{"source": "Audio Downloader", "target": "Audio Splicer"}, {"source": "Audio Splicer", "target": "Audio-to-Text"}, {"source": "Audio-to-Text", "target": "Text Grammar Checker"}]}` |
| Incorrect Tool Dependency | I want to create a more engaging version of this short text: 'Join us for a fun-filled evening!' and find some videos related to its sentiment. `{"task_nodes": [{"task": "Article Spinner", "arguments": ["<node-2>"]}, {"task": "Text Expander", "arguments": ["Join us for a fun-filled evening!"]}, {"task": "Text Sentiment Analysis", "arguments": ["<node-1>"]}, {"task": "Video Search", "arguments": ["<node-2>"]}], "task_links": [{"source": "Text Expander", "target": "Text Sentiment Analysis"}, {"source": "Text Sentiment Analysis", "target": "Article Spinner"}, {"source": "Text Sentiment Analysis", "target": "Video Search"}]}` |

- gpt-3.5-turbo and code-llama-13b show a progressively higher number of errors. Notably, the 'Incorrect Tool Dependency' is the most common across all models, highlighting the challenge LLMs face in predicting accurate parameters for tools.

Table 9: Error Distribution in Different LLMs.

|  | Required Tool Missing | Tool Dependency Error | Tool Parameter Error |
|---|---|---|---|
| gpt-4 | 0 | 2 | 3 |
| gpt-3.5-turbo | 2 | 8 | 11 |
| code-llama-13b | 4 | 9 | 13 |

Further, we present specific cases in Table 10 to elucidate the nature of prediction errors in these LLMs. Given the following example, gpt-4 correctly interpreted the task in the given example, underscoring its advanced task automation capabilities. Conversely, gpt-3.5-turbo and code-llama-13b omitted a critical tool ('Audio Downloader'), resulting in a 'Missing Required Tool' error. Additionally, code-llama-13b encountered compounded errors, including 'Tool Parameter Error' and 'Incorrect Tool Dependency'.

```
Instruction:
I need an audio file downloaded from 'https://www.example.com/
example.wav', then please reduce the background noise and apply a
reverb effect according to my instruction 'reverb 50\%'. Finally,
combine it with the audio file 'example.wav'.
Gold tool invocation graph:
"task_nodes": [
{"task": "Audio Downloader", "arguments": ["https://www.example.
com/example.wav"]},
{"task": "Audio Noise Reduction", "arguments": ["<node-0>"]},
{"task": "Audio Effects", "arguments": ["<node-1>", "reverb
50%"]},
{"task": "Audio Splicer", "arguments": ["<node-2>", "example.wav
"]}]
```

## A.5 METRICS FOR RANKING CONSISTENCY

To compute the consistency of two rankings where the number of observations is n, we introduce two correlation coefficients: Spearman's $\rho$ and Kendall's $\tau$. In our work, they refer to the human and TaskBench rankings of large language models in terms of task automation capabilities.

**Spearman's** $\rho$ measures the strength and direction of the rank association between two variables. To calculate Spearman's $\rho$, start by assigning ranks to each observation in both sets of data. For any tied values, assign the average rank. Next, compute the difference in ranks between the two datasets for each observation, and then square these differences. The coefficient is calculated as follow:

$$\rho = 1 - \frac{6 \times \text{ sum of squared rank differences}}{n\left(n^2 - 1\right)} \tag{4}$$

**Kendall's** $\tau$ is calculated based on the consistency and inconsistency of pairs between two rankings. For both rankings, we will consider all possible pairs of items in them. For each pair of items, if the relative position is correct in both rankings, then we call this a "consistent pair". If the relative position is wrong, then we call this an "inconsistent pair".

$$\tau = \frac{(\text{number of concordant pairs }) - (\text{ number of discordant pairs })}{n(n-1)/2} \tag{5}$$

## A.6 ANALYSIS

### A.6.1 DIFFERENT NUMBER OF TOOLS

Table 10: Case Study of Prediction Errors in LLMs.

| LLM | Predicted Tool Invocation Graph | Error Type |
|---|---|---|
| gpt-4 | `{"task_nodes": [{"task": "Audio Downloader", "arguments": ["https://www.example.com/example.wav"]}, {"task": "Audio Noise Reduction", "arguments": ["<node-0>"]}, {"task": "Audio Effects", "arguments": ["<node-1>", "reverb 50%"]}, {"task": "Audio Splicer", "arguments": ["<node-2>", "example.wav"]}]}` | None |
| gpt-3.5-turbo | `{"task_nodes": [{"task": "Audio Noise Reduction", "arguments": ["""https://www.example.com/example.wav"""]}, {"task": "Audio Effects", "arguments": ["<node-0>", "reverb 50%"]}, {"task": "Audio Splicer", "arguments": ["<node-1>", "example.wav"]}]}` | Missing Required Tool |
| code-llama-13b | `{"task_nodes": [{"task": "Audio Noise Reduction", "arguments": ["example.wav"]}, {"task": "Audio Effects", "arguments": ["<node-0>", "reverb 50%"]}, {"task": "Audio Splicer", "arguments": ["<node-1>", "<node-0>"]}]}` | Missing Required Tool
Tool Parameter Error
Incorrect Tool Dependency |

The greater the number of tool nodes the tool graphs contain, the more challenging it is for LLM to perform task automation. To make a clear understanding of the correlation between the number of nodes in the tool graph and the performance of LLMs in task automation, we conduct a detailed statistical analysis in Table 11. This analysis includes various metrics such as node-wise F1, node set accuracy, edge-wise F1, edge set accuracy, and graph accuracy, which measure the exact-match accuracy of the node set, edge set, and the entire graph, respectively.

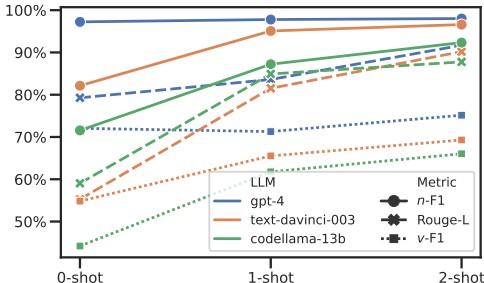

Figure 4: Performance in the few-shot setting for the Daily Life APIs domain. *R-L* for task decomposition; *n-F1* for tool invocation; and *v-F1* for parameter prediction.

From Table 11, we observed that as the number of tools in the tool graphs increases, there is a clear downward trend in various performance metrics such as node set accuracy, edge set accuracy, and graph accuracy. This trend confirms that tool graphs with a higher number of tools present more complex challenges for LLMs in task automation. Specifically, the result shows a significant drop in performance metrics when moving from simpler graphs (1-2 tools) to more complex ones (3 or more tools). For instance, while single-node graphs achieve a high graph accuracy of 96.16%, this metric falls to 39.31% for graphs with 6 tools and further decreases to 25.00% for 8-node graphs.

This correlation between the number of tools and the difficulty of the test cases can be attributed to the increased complexity in understanding and processing more extensive and intricate links between

Table 11: Task automation performance with different number of tools on GPT-4

| # tool nodes | supports | node set accuracy | edge set accuracy | graph accuracy |
|---|---|---|---|---|
| 1 (single) | 2,059 | 96.16 | - | 96.16 |
| 2 | 278 | 86.33 | 84.53 | 84.53 |
| 3 | 1,313 | 67.93 | 60.70 | 60.39 |
| 4 | 1,280 | 64.29 | 75.62 | 54.37 |
| 5 | 731 | 54.03 | 70.53 | 41.58 |
| 6 | 290 | 50.34 | 39.31 | 39.31 |
| 7 | 151 | 49.66 | 36.42 | 36.42 |
| 8 | 60 | 35.00 | 25.00 | 25.00 |
| 9 | 55 | 38.18 | 21.81 | 21.81 |
| 10 | 64 | 39.06 | 31.25 | 31.25 |
| overall | 6,281 | 73.52 | 67.55 | 67.25 |

tools. As the number of tools grows, LLMs must handle a larger set of possible dependencies, which significantly challenges their predictive and analytical capabilities. The results from this analysis underline the importance of continuous advancements in LLM capabilities to keep pace with the increasing complexity of tasks in various domains.

### A.6.2 FEW-SHOT SETTING

In-context learning is a crucial capability for LLMs, that can improve the performance of LLMs by providing a few examples. In TASKBENCH, we also provide a fixed number of demonstrations in the designed prompt to advance the capability of LLMs in automation. Therefore, we also investigate the effect of the number of demonstrations in our setting. The results are reported in Table 12 and Figure 4. We can find that as the number of demonstrations increases, it can receive significant improvements of LLMs at different dimensions (e.g., task decomposition, tool invocation, and parameter prediction). For example, codellama-13b with a 2-shot setting can obtain 20.78% and 21.82% improvements to the zero-shot setting in *n-F1* and *v-F1*. These results underscore the effect of the demonstrations in improving LLMs for task automation.

### A.7 DETAIL COMPARISON OF LLMS

We present the performance of more open-source large language models (Team, 2023a; Li et al., 2023a; Team, 2023b) on TASKBENCH. The performance metrics for task decomposition, tool invocation, and parameter prediction are shown in Table 13, Table 14, and Table 15, respectively.

### A.8 PROMPT FOR BACK-INSTRUCT DATA ENGINE

We utilize the following prompt template for the "Back-Instruct" Data Engine. Each sample is generated through a single prompting. We assign "instruction", "tool invocation graph", and "self-critics" to specific fields in the prompt, and then populate the relevant fields to complete the data generation in a single prompt.

```
Given a tool graph where tools serve as nodes and invoking chains between
 tools act as edges, the following tools (nodes) are available with their
 respective descriptions and input/output types:
{NODES}

These tools can be connected as follows, where the directed edges
represent invoking chains between tools:
{EDGES}

Based on the above tool graph, please skillfully generate the
corresponding task steps, user request, and tool invoking graph.

Requirements:
1. The generated user request should be clear, self-contained (with user-
specified text, image, video, audio, and content included within the
```

Table 12: Performance in the few-shot setting. Rouge-L (*R-L*) reflects the performance on task decomposition; Node F1 (*n-F1*) and Edge F1 (*e-F1*) indicate the performance on tool invocation; and Parameter Name F1 (*t-F1*) and Parameter Name & Value F1 (*v-F1*) indicate the performance on parameter prediction.

| Shot | LLM | R-L ↑ | n-F1 ↑ | e-F1 ↑ | t-F1 ↑ | v-F1 ↑ |
|---|---|---|---|---|---|---|
| 0-shot | gpt-4 | 79.27 | 97.23 | 34.52 | 97.11 | 72.05 |
| | gpt-3.5-turbo | 48.72 | 86.45 | 51.77 | 83.85 | 56.34 |
| | text-davinci-003 | 55.28 | 82.13 | 50.75 | 80.63 | 54.83 |
| | codellama-13b | 59.05 | 71.55 | 36.57 | 63.04 | 44.21 |
| | wizardlm-13b | 40.58 | 65.39 | 20.87 | 55.96 | 38.56 |
| | vicuna-13b-v1.5 | 48.34 | 68.32 | 26.73 | 51.47 | 35.71 |
| | nous-hermes-13b | 36.58 | 50.44 | 13.96 | 36.32 | 25.07 |
| 1-shot | gpt-4 | 83.60 | 97.78 | 50.56 | 97.82 | 71.28 |
| | gpt-3.5-turbo | 76.16 | 90.87 | 51.14 | 90.18 | 62.31 |
| | text-davinci-003 | 81.52 | 95.08 | 72.00 | 94.73 | 65.52 |
| | codellama-13b | 84.91 | 87.21 | 54.89 | 83.71 | 61.76 |
| | vicuna-13b-v1.5 | 75.52 | 73.96 | 11.17 | 62.39 | 45.81 |
| | nous-hermes-13b | 73.04 | 72.69 | 2.76 | 63.77 | 46.48 |
| | wizardlm-13b | 75.96 | 67.93 | 12.26 | 53.59 | 39.08 |
| 2-shot | gpt-4 | 91.69 | 98.02 | 52.49 | 97.94 | 75.15 |
| | gpt-3.5-turbo | 89.07 | 96.03 | 39.72 | 95.28 | 69.04 |
| | text-davinci-003 | 90.18 | 96.59 | 72.37 | 96.19 | 69.29 |
| | codellama-13b | 87.76 | 92.33 | 64.17 | 88.60 | 66.03 |
| | nous-hermes-13b | 78.88 | 82.42 | 43.55 | 74.68 | 54.39 |
| | wizardlm-13b | 80.20 | 79.25 | 33.51 | 70.69 | 52.20 |
| | vicuna-13b-v1.5 | 79.67 | 79.84 | 37.50 | 71.45 | 51.82 |

Table 13: Evaluation for task decomposition. The scores for Rouge-1 (*R1*), Rouge-2 (*R2*), and BertScore F1 (*BsF*) for the generated step descriptions in comparison to the ground truth steps.

| TASK DECOMPOSITION TASK - Step-by-step task decomposition | | | | | | | | | |
|---|---|---|---|---|---|---|---|---|---|
| | Hugging Face Tools | | | Multimedia Tools | | | Daily Life APIs | | |
| LLM | R1 ↑ | R2 ↑ | BsF ↑ | R1 ↑ | R2 ↑ | BsF ↑ | R1 ↑ | R2 ↑ | BsF ↑ |
| gpt-4 | 52.56 | 30.49 | 90.12 | 60.92 | 40.02 | 91.17 | 86.74 | 75.06 | 97.52 |
| gpt-3.5-turbo | 43.31 | 21.83 | 88.48 | 50.08 | 28.74 | 89.54 | 79.54 | 67.00 | 96.33 |
| text-davinci-003 | 37.04 | 17.87 | 87.04 | 49.53 | 28.08 | 89.20 | 84.58 | 72.17 | 97.15 |
| claude-2 | 44.49 | 21.37 | 88.71 | 49.19 | 23.71 | 89.22 | 81.28 | 68.06 | 95.65 |
| codellama-13b | 39.08 | 18.62 | 88.32 | 44.95 | 23.57 | 88.68 | 89.30 | 82.37 | 97.75 |
| wizardlm-13b | 34.77 | 15.58 | 87.37 | 36.34 | 17.80 | 87.29 | 81.14 | 70.90 | 96.12 |
| vicuna-13b-v1.5 | 37.48 | 17.25 | 87.90 | 45.07 | 23.89 | 88.91 | 81.07 | 70.55 | 96.12 |
| nous-hermes-13b | 37.71 | 17.15 | 88.18 | 36.26 | 16.40 | 87.53 | 78.76 | 67.91 | 95.59 |
| codellama-7b | 39.45 | 18.97 | 88.51 | 44.29 | 23.26 | 88.85 | 58.40 | 39.72 | 91.41 |
| baichuan-13b-chat | 19.99 | 6.01 | 83.83 | 20.37 | 3.71 | 83.21 | 50.62 | 27.84 | 88.39 |
| llama-2-13b-chat | 39.77 | 18.93 | 88.69 | 26.09 | 7.76 | 84.73 | 46.80 | 23.21 | 87.82 |
| vicuna-7b-v1.5 | 27.58 | 10.26 | 85.64 | 40.02 | 20.13 | 88.54 | 41.23 | 21.54 | 87.29 |
| mpt-7b-chat | 33.32 | 12.76 | 87.18 | 30.98 | 11.88 | 86.00 | 45.57 | 21.41 | 87.17 |
| llama-2-7b-chat | 24.59 | 8.94 | 85.45 | 35.07 | 16.16 | 87.59 | 39.29 | 17.45 | 86.45 |
| internlm-chat-7b | 20.68 | 7.24 | 83.70 | 16.40 | 3.52 | 82.81 | 42.81 | 20.74 | 85.98 |
| longchat-7b-v1.5 | 27.29 | 9.14 | 85.49 | 38.22 | 18.36 | 87.65 | 29.60 | 15.05 | 83.78 |

Table 14: Evaluation for tool invocation. Node F1 *(n-F1)* for node prediction and Edge F1 *(e-F1)* for edge prediction. For nodes, a prediction is deemed positive if the predicted node's ID aligns with any of the ground-truth node labels. For edges, both the source and target nodes of a predicted edge must correspond exactly. Normalized Edit Distance *(NED)* measures the normalized number of operations required to correct the prediction for chain structure.

| | LLM | Node | Chain | | | DAG | | Overall | |
|---|---|---|---|---|---|---|---|---|---|
| | | *n-F1* ↑ | *n-F1* ↑ | *e-F1* ↑ | *NED* ↓ | *n-F1* ↑ | *e-F1* ↑ | *n-F1* ↑ | *e-F1* ↑ |
| **Hugging Face Tools** | gpt-4 | 84.34 | 80.35 | 56.78 | 43.95 | 82.14 | 58.13 | 81.37 | 56.18 |
| | gpt-3.5-turbo | 56.89 | 72.15 | 40.74 | 50.03 | 74.22 | 42.02 | 69.81 | 35.31 |
| | text-davinci-003 | 40.65 | 66.84 | 36.98 | 51.53 | 66.00 | 37.27 | 60.44 | 31.16 |
| | claude-2 | 69.77 | 80.07 | 48.61 | 41.78 | 83.17 | 52.66 | 79.15 | 45.19 |
| | codellama-13b | 43.65 | 55.86 | 18.70 | 64.10 | 54.90 | 16.39 | 53.56 | 15.60 |
| | baichuan-13b-chat | 58.34 | 52.88 | 7.66 | 63.10 | 52.51 | 8.33 | 53.69 | 7.52 |
| | nous-hermes-13b | 58.67 | 52.04 | 8.74 | 64.52 | 53.13 | 7.45 | 53.51 | 8.02 |
| | llama-2-13b-chat | 43.58 | 50.14 | 8.39 | 66.44 | 50.05 | 8.30 | 48.82 | 7.49 |
| | vicuna-13b-v1.5 | 51.76 | 50.15 | 8.02 | 68.68 | 52.53 | 9.54 | 51.03 | 7.43 |
| | codellama-7b | 18.80 | 48.60 | 8.85 | 65.26 | 47.30 | 8.96 | 38.80 | 5.96 |
| | vicuna-33b-v1.3 | 42.72 | 41.83 | 5.37 | 73.71 | 45.67 | 6.71 | 42.96 | 4.91 |
| | vicuna-7b-v1.5 | 36.20 | 45.15 | 2.91 | 70.48 | 44.44 | 2.79 | 43.23 | 2.60 |
| | wizardlm-13b | 54.67 | 54.46 | 1.98 | 62.55 | 54.90 | 1.84 | 54.60 | 1.91 |
| | llama-2-7b-chat | 14.88 | 34.08 | 0.74 | 81.06 | 31.48 | 0.77 | 28.25 | 0.72 |
| | longchat-7b-v1.5 | 44.95 | 49.41 | 0.32 | 67.26 | 49.06 | 1.04 | 48.48 | 0.51 |
| | internlm-chat-7b | 33.95 | 22.94 | 0.72 | 85.96 | 21.80 | 1.02 | 24.09 | 0.79 |
| | mpt-7b-chat | 15.67 | 22.38 | 0.10 | 88.84 | 22.97 | 0.14 | 21.38 | 0.11 |
| **Multimedia Tools** | gpt-4 | 97.13 | 88.92 | 71.84 | 35.15 | 90.49 | 72.72 | 90.27 | 71.88 |
| | gpt-3.5-turbo | 53.55 | 76.91 | 52.91 | 43.79 | 76.89 | 51.44 | 73.23 | 47.07 |
| | text-davinci-003 | 59.15 | 77.58 | 54.53 | 43.03 | 75.59 | 50.58 | 74.17 | 48.96 |
| | claude-2 | 66.15 | 82.69 | 60.76 | 37.04 | 85.47 | 63.42 | 81.07 | 56.62 |
| | codellama-13b | 43.70 | 66.44 | 28.84 | 50.60 | 67.36 | 27.94 | 62.98 | 25.12 |
| | vicuna-13b-v1.5 | 66.64 | 58.04 | 15.94 | 57.73 | 60.50 | 16.56 | 59.96 | 14.97 |
| | codellama-7b | 40.43 | 55.67 | 16.89 | 57.92 | 57.14 | 17.11 | 53.50 | 15.20 |
| | nous-hermes-13b | 60.58 | 57.99 | 8.46 | 59.02 | 58.52 | 9.16 | 58.53 | 8.25 |
| | baichuan-13b-chat | 45.59 | 41.69 | 4.61 | 66.09 | 41.22 | 6.08 | 42.06 | 4.96 |
| | longchat-7b-v1.5 | 43.54 | 42.51 | 4.18 | 68.69 | 43.14 | 4.36 | 42.85 | 3.91 |
| | wizardlm-13b | 55.13 | 50.55 | 4.12 | 61.28 | 50.01 | 5.41 | 51.09 | 4.39 |
| | vicuna-7b-v1.5 | 36.22 | 48.43 | 4.46 | 65.37 | 47.08 | 4.05 | 46.06 | 4.00 |
| | llama-2-13b-chat | 38.02 | 45.36 | 1.43 | 67.23 | 45.44 | 1.83 | 44.16 | 1.52 |
| | llama-2-7b-chat | 16.49 | 30.90 | 0.74 | 77.07 | 29.25 | 0.90 | 27.17 | 0.75 |
| | internlm-chat-7b | 36.39 | 20.77 | 1.03 | 85.40 | 22.16 | 1.19 | 22.48 | 1.07 |
| | vicuna-33b-v1.3 | 27.53 | 2.33 | 0.04 | 98.01 | 2.73 | 0.00 | 5.66 | 0.02 |
| | mpt-7b-chat | 11.85 | 7.67 | 0.18 | 96.22 | 8.55 | 0.32 | 8.44 | 0.22 |
| **Daily Life APIs** | gpt-4 | 96.92 | 98.05 | 78.42 | 39.99 | 97.47 | 38.84 | 97.78 | 70.56 |
| | gpt-3.5-turbo | 72.47 | 92.90 | 61.28 | 42.61 | 91.83 | 37.46 | 90.87 | 51.14 |
| | text-davinci-003 | 92.64 | 95.79 | 66.37 | 40.70 | 94.34 | 38.45 | 95.08 | 52.00 |
| | claude-2 | 79.73 | 95.62 | 62.17 | 35.39 | 93.76 | 55.96 | 93.73 | 57.66 |
| | codellama-13b | 89.75 | 87.85 | 66.19 | 45.43 | 85.68 | 36.94 | 87.22 | 55.06 |
| | codellama-7b | 40.19 | 63.46 | 32.24 | 59.63 | 62.33 | 17.96 | 61.22 | 25.27 |
| | llama-2-13b-chat | 34.11 | 58.21 | 20.56 | 67.24 | 56.35 | 11.70 | 56.26 | 15.73 |
| | vicuna-33b-v1.3 | 30.25 | 58.03 | 20.39 | 63.14 | 55.34 | 11.09 | 53.90 | 15.31 |
| | longchat-7b-v1.5 | 34.20 | 48.07 | 16.99 | 71.62 | 53.27 | 13.37 | 47.76 | 13.52 |
| | vicuna-7b-v1.5 | 46.51 | 53.83 | 17.49 | 66.21 | 52.30 | 10.75 | 52.51 | 13.22 |
| | wizardlm-13b | 92.27 | 65.81 | 14.84 | 56.51 | 63.64 | 7.90 | 67.86 | 12.29 |
| | vicuna-13b-v1.5 | 90.59 | 72.92 | 12.96 | 52.81 | 70.61 | 8.82 | 73.96 | 11.17 |
| | baichuan-13b-chat | 52.50 | 51.56 | 11.05 | 70.36 | 53.74 | 7.99 | 52.41 | 9.51 |
| | internlm-chat-7b | 33.08 | 28.23 | 6.75 | 86.40 | 29.79 | 4.55 | 29.00 | 5.79 |
| | llama-2-7b-chat | 20.11 | 31.75 | 5.66 | 83.97 | 32.78 | 3.39 | 31.05 | 4.16 |
| | nous-hermes-13b | 92.50 | 71.05 | 2.87 | 54.75 | 70.71 | 2.55 | 72.64 | 2.75 |
| | mpt-7b-chat | 14.75 | 17.29 | 2.83 | 92.75 | 16.10 | 1.85 | 16.51 | 1.96 |

Table 15: Evaluation for tool parameter prediction. Parameter Name F1 *(t-F1)* evaluates (task, parameter name) pairs, while Parameter Name & Value F1 *(v-F1)* assesses (task, parameter name, parameter value) triples.

| | **TOOL PARAMETER PREDICTION TASK - Predicts parameters for the tool execution.** | | | | | | | |
| | | **Node** | | **Chain** | | **DAG** | | **Overall** | |
| | **LLM** | *t-F1*↑ | *v-F1*↑ | *t-F1* ↑ | *v-F1*↑ | *t-F1* ↑ | *v-F1*↑ | *t-F1* ↑ | *v-F1*↑ |
|---|---|---|---|---|---|---|---|---|---|
| **Hugging Face Tools** | gpt-4 | 79.87 | 74.36 | 77.42 | 58.79 | 78.96 | 60.37 | 78.15 | 61.40 |
| | gpt-3.5-turbo | 37.39 | 19.67 | 61.76 | 41.46 | 63.33 | 44.02 | 57.30 | 37.45 |
| | text-davinci-003 | 38.30 | 27.52 | 58.18 | 39.36 | 58.76 | 40.03 | 54.01 | 36.92 |
| | claude-2 | 48.29 | 32.33 | 67.10 | 45.81 | 69.91 | 47.93 | 64.52 | 43.87 |
| | codellama-13b | 19.96 | 12.54 | 37.21 | 21.61 | 35.31 | 21.06 | 32.82 | 19.23 |
| | nous-hermes-13b | 46.22 | 31.06 | 35.34 | 13.18 | 36.53 | 13.51 | 37.60 | 16.89 |
| | llama-2-13b-chat | 29.62 | 20.61 | 32.33 | 13.30 | 33.14 | 14.22 | 31.97 | 15.12 |
| | wizardlm-13b | 43.80 | 26.01 | 37.16 | 11.56 | 39.33 | 13.12 | 38.89 | 14.63 |
| | baichuan-13b-chat | 46.05 | 29.54 | 30.59 | 9.38 | 30.75 | 9.63 | 33.16 | 13.02 |
| | longchat-7b-v1.5 | 34.81 | 19.37 | 33.20 | 10.71 | 33.84 | 12.38 | 33.68 | 13.45 |
| | vicuna-13b-v1.5 | 25.59 | 13.16 | 29.09 | 10.78 | 30.97 | 12.62 | 28.81 | 11.76 |
| | vicuna-33b-v1.3 | 20.67 | 11.93 | 22.68 | 8.69 | 25.08 | 10.64 | 22.83 | 9.90 |
| | vicuna-7b-v1.5 | 20.78 | 12.61 | 25.94 | 9.47 | 26.44 | 10.53 | 24.92 | 10.47 |
| | codellama-7b | 13.20 | 4.47 | 27.78 | 12.18 | 27.39 | 12.55 | 23.15 | 9.57 |
| | internlm-chat-7b | 20.42 | 13.99 | 14.17 | 4.23 | 14.50 | 5.46 | 15.25 | 6.24 |
| | llama-2-7b-chat | 7.56 | 2.49 | 16.68 | 2.87 | 15.15 | 3.26 | 13.69 | 2.84 |
| | mpt-7b-chat | 6.26 | 1.73 | 11.16 | 1.65 | 11.76 | 2.26 | 10.15 | 1.81 |
| **Multimedia Tools** | gpt-4 | 95.58 | 87.80 | 86.88 | 72.86 | 88.43 | 73.44 | 88.27 | 74.65 |
| | gpt-3.5-turbo | 44.74 | 11.70 | 72.13 | 50.77 | 72.11 | 48.84 | 67.65 | 43.53 |
| | text-davinci-003 | 60.38 | 20.44 | 71.77 | 49.11 | 70.12 | 45.07 | 69.58 | 43.59 |
| | claude-2 | 53.57 | 23.79 | 75.83 | 59.59 | 77.70 | 60.61 | 73.08 | 54.40 |
| | codellama-13b | 31.98 | 16.00 | 51.93 | 32.72 | 53.06 | 32.23 | 48.60 | 29.43 |
| | codellama-7b | 31.84 | 23.06 | 39.38 | 24.54 | 40.91 | 25.07 | 38.51 | 24.42 |
| | vicuna-13b-v1.5 | 52.71 | 35.78 | 37.87 | 19.88 | 41.15 | 22.17 | 40.95 | 23.00 |
| | nous-hermes-13b | 50.13 | 38.10 | 41.49 | 16.98 | 42.70 | 18.06 | 43.14 | 20.62 |
| | wizardlm-13b | 49.84 | 33.85 | 36.11 | 13.85 | 37.12 | 15.88 | 38.50 | 17.69 |
| | longchat-7b-v1.5 | 31.00 | 21.37 | 26.84 | 10.84 | 27.88 | 11.65 | 27.82 | 12.97 |
| | vicuna-7b-v1.5 | 28.83 | 17.89 | 29.23 | 11.83 | 30.63 | 12.85 | 29.55 | 13.22 |
| | baichuan-13b-chat | 40.64 | 28.19 | 25.13 | 8.15 | 25.43 | 8.78 | 27.29 | 11.14 |
| | llama-2-13b-chat | 28.61 | 17.19 | 30.03 | 8.97 | 30.62 | 10.00 | 29.95 | 10.73 |
| | internlm-chat-7b | 24.20 | 16.72 | 11.40 | 4.30 | 12.97 | 5.30 | 13.04 | 5.75 |
| | llama-2-7b-chat | 14.06 | 7.13 | 20.13 | 5.09 | 19.70 | 5.54 | 18.67 | 5.67 |
| | mpt-7b-chat | 4.08 | 2.17 | 2.74 | 0.62 | 3.20 | 0.85 | 3.10 | 0.99 |
| | vicuna-33b-v1.3 | 9.43 | 4.75 | 0.74 | 0.14 | 0.83 | 0.13 | 2.21 | 0.98 |
| **Daily Life APIs** | gpt-4 | 96.84 | 77.43 | 98.21 | 71.30 | 97.31 | 70.03 | 97.82 | 71.28 |
| | gpt-3.5-turbo | 69.65 | 52.77 | 92.66 | 63.80 | 91.06 | 62.16 | 90.18 | 62.31 |
| | text-davinci-003 | 91.49 | 69.91 | 95.66 | 65.85 | 93.80 | 64.06 | 94.73 | 65.52 |
| | claude-2 | 78.14 | 60.10 | 95.13 | 64.95 | 92.49 | 64.69 | 92.89 | 64.47 |
| | codellama-13b | 86.37 | 72.00 | 84.27 | 61.48 | 82.26 | 59.85 | 83.70 | 61.65 |
| | vicuna-13b-v1.5 | 83.68 | 68.39 | 61.35 | 44.28 | 58.28 | 42.08 | 62.39 | 45.81 |
| | nous-hermes-13b | 79.79 | 63.82 | 62.05 | 44.85 | 62.85 | 45.19 | 63.70 | 46.42 |
| | wizardlm-13b | 89.38 | 73.85 | 50.93 | 36.28 | 48.91 | 35.09 | 53.50 | 38.99 |
| | codellama-7b | 31.63 | 21.10 | 57.68 | 38.18 | 56.69 | 36.97 | 54.99 | 36.11 |
| | vicuna-33b-v1.3 | 21.17 | 17.18 | 45.33 | 33.03 | 42.44 | 31.30 | 41.28 | 30.42 |
| | vicuna-7b-v1.5 | 27.72 | 19.77 | 38.47 | 25.97 | 37.26 | 25.15 | 36.64 | 24.87 |
| | llama-2-13b-chat | 10.40 | 7.32 | 39.82 | 25.82 | 38.38 | 24.80 | 36.61 | 23.75 |
| | baichuan-13b-chat | 32.49 | 21.66 | 37.58 | 23.69 | 39.39 | 23.58 | 37.75 | 23.46 |
| | longchat-7b-v1.5 | 14.99 | 12.12 | 26.88 | 18.49 | 32.22 | 22.11 | 26.60 | 18.56 |
| | internlm-chat-7b | 18.67 | 15.22 | 18.59 | 12.74 | 19.21 | 14.07 | 18.83 | 13.35 |
| | llama-2-7b-chat | 6.55 | 4.12 | 16.89 | 10.73 | 18.64 | 11.38 | 16.03 | 10.02 |
| | mpt-7b-chat | 2.80 | 2.04 | 7.00 | 4.55 | 6.40 | 3.94 | 5.82 | 3.76 |

```
request) and practical (designed to help users solve a tangible problem).
 The task steps must strictly adhere to the tool graph (nodes and edges)
and be reasonable. The tool invoking graph must align with both the task
steps and the provided tool graph.
2. The user request should be decomposable into task steps that the tool
invoking graph can solve.
3. Each task step must correspond to a tool node in both the tool graph
and the tool invoking graph. The number of task steps must equal the
number of nodes. Each tool node can only be used once.
4. If the user request requires image/audio/video resources, use files
named 'example.[jpg/mp4/wav/png]'.
5. The dependencies among task steps must be consistent with the edges of
 both the tool graph and the tool invoking graph.

Now, please generate your result (with random seed {seed}) in a strict
JSON format:

{
"task_steps": [step description for one or more steps],
"user_request": "your high-quality and self-contained synthesized request
",
"invoking_graph": {
    "nodes": [
        {
            "id": "tool name",
            "input": [either user-specified text or resource file '
            example.[jpg/mp4/wav/png]' from the user request, or the name
             of the dependent tool whose output this node requires]
        }
    ],
    "links": [{"source": "tool name i", "target": "tool name j"}]
    },
"check_by_teacher": "This field is filled by your strict and well-trained
 teacher, minor mistakes are complete intolerable to him. He evaluated
whether your synthesized user request, tool invoking graph are valid and
whether they are aligned with the given tool graph (strictly checked step
 by step according to the above requirements). Some comments from him
place here (start with 'Let me check your result step by step, and
evaluate the 'Executable' and 'Correct' of the tool invoking graph (
Executable means that the tool invoking graph executed successfully,
regardless of alignment with the given tool graph. While Correct implies
that the tool invoking graph are not only 'Executable' but also strictly
consistent (with strictly same nodes and same edges) with the given tool
graph). After carefully evaluating, found some mistakes:' and end with a
conclusion: 'Conclusion: Executable: no/yes, Correct: no/yes'.)
}:
```

## A.9  PROMPT FOR INFERENCE

For a fair and standardized evaluation, we provide prompt templates for inference.

```
# Tools List #
FOR tool in {tool_list}:
    {tool["id"]: {tool["description"]}}
    Parameters: {tool["parameter"]}
END FOR

# GOAL #:
Based on the above tools, I want you to generate task steps and a task
graph (tool invocation graph, including nodes and edges) to address the #
 USER REQUEST #. The format must be in strict JSON format, like:
{
    "task_steps": [step description for one or more steps],
    "task_nodes": [{
```

```
        "task": "tool name must be from # TOOL LIST #",          "
        arguments": [a concise list of arguments for the tool. This can
        be original text, a user-mentioned filename, or the tag '<node-j
        >' (starting from 0) to refer to the output of the j-th node.]
        }]
    "task_links": [{"source": "task name i", "target": "task name j"}],
}.

# REQUIREMENTS #:
1. The generated task steps and task nodes must resolve the given user
request # USER REQUEST # perfectly. Task names must be selected from #
TASK LIST #.
2. The task steps should align strictly with the task nodes, and the
number of task steps should be the same as the task nodes.
3. The dependencies among task steps should be consistent with the
argument dependencies of the task nodes.
4. The tool arguments should align with the parameter field of # TASK
LIST #.

# EXAMPLE #:
FOR demo IN {demos}:
# USER REQUEST #:
{demo["user_request"]}
# RESULT #:
{(demo["result"])}
END FOR

# USER REQUEST #:
{{user_request}}
Now, please generate your result in strict JSON format:
# RESULT #:
```

### A.10    TOOLS IN THE TOOL GRAPH

We show some of the tools used in the construction of the tool graph, including the tool name, tool description and parameters of the tool. In the Daily Life APIs domain, we resorted to manual construction because of the scarcity of publicly available APIs. We crafted 40 APIs tailored to common daily life activities such as shopping, education, and travel. Our focus is solely on producing the API documentation without implementing the actual functionality. Some of the tools on the Hugging Face tools, Multimedia tools and Daily Life APIs domains are shown in Table 16, Table 17, and Table 18, respectively.

In order to visualize the complete tool graph we constructed, we take the Multimedia domain as an example to render the tool graph with resource dependencies. As shown in Figure 5, nodes in the graph denote tools, and directed edges indicate that the output type of the source tool matches the input type of the target tool.

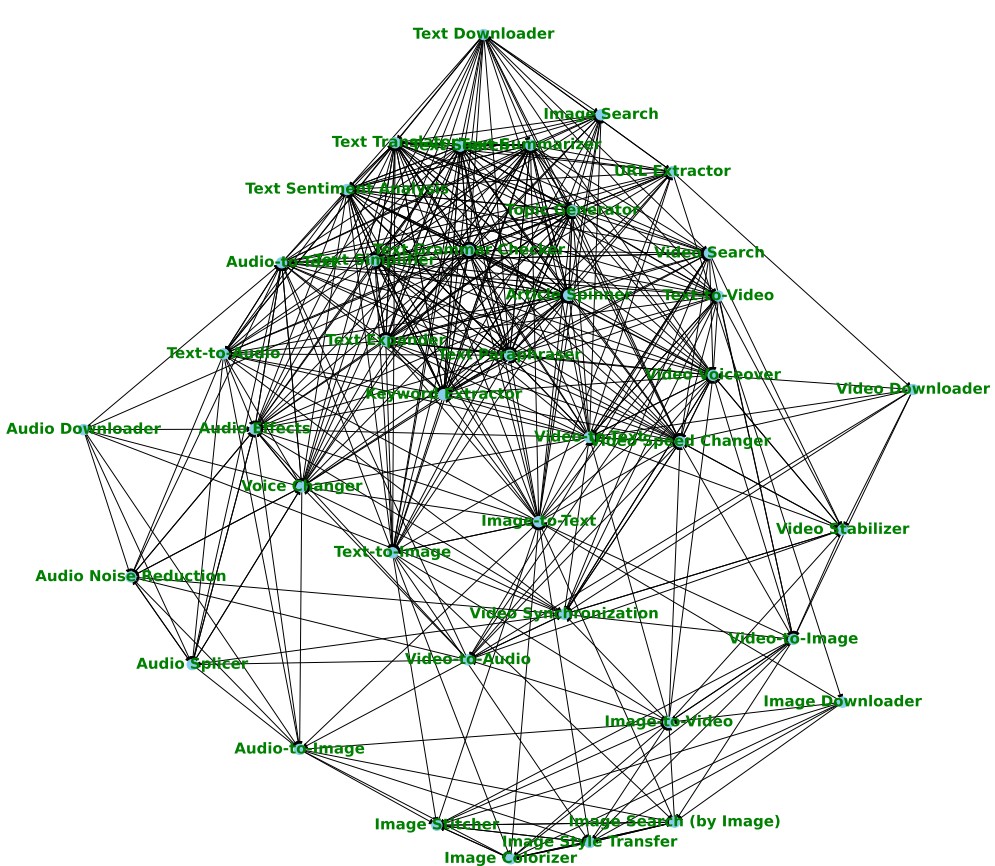

Figure 5: Constructed tool graph with resource dependencies on the Multimedia domain.

Table 16: Hugging Face tools and their descriptions

| Name | Description | Parameters |
|---|---|---|
| Translation | Translation is the task of converting text from one language to another. | ['text'] |
| Summarization | Summarization is the task of producing a shorter version of a document while preserving its important information. Some models can extract text from the original input, while other models can generate entirely new text. | ['text'] |
| Question Answering | Question Answering models can retrieve the answer to a question from a given text, which is useful for searching for an answer in a document. | ['text', 'text'] |
| Text Generation | Generating text is the task of producing new text. These models can, for example, fill in incomplete text or paraphrase. | ['text'] |
| Object Detection | Object Detection models allow users to identify objects of certain defined classes. Object detection models receive an image as input and output the images with bounding boxes and labels on detected objects. | ['image'] |
| Image Classification | Image classification is the task of assigning a label or class to an entire image. Images are expected to have only one class for each image. Image classification models take an image as input and return a prediction about which class the image belongs to. | ['image'] |
| Image-to-Image | Image-to-image is the task of transforming a source image to match the characteristics of a target image or a target image domain. Any image manipulation and enhancement is possible with image to image models. | ['image'] |
| Image-to-Text | Image to text models output a text from a given image. Image captioning or optical character recognition can be considered as the most common applications of image to text. | ['image'] |
| Text-to-Image | Generates images from input text. These models can be used to generate images based on text prompts. | ['text'] |
| Text-to-Video | Generates videos from input text. These models can be used to generate videos based on text prompts. | ['text'] |
| Visual Question Answering | Visual Question Answering is the task of answering questions based on an image. | ['image', 'text'] |
| Image Segmentation | Image Segmentation divides an image into segments where each pixel in the image is mapped to an object. This task has multiple variants such as instance segmentation, panoptic segmentation and semantic segmentation. | ['image'] |
| Depth Estimation | Depth estimation is the task of predicting depth of the objects present in an image. | ['image'] |
| Text-to-Speech | Text-to-Speech (TTS) is the task of generating natural sounding speech given text input. TTS models can be extended to have a single model that generates speech for multiple speakers and multiple languages. | ['text'] |
| Automatic Speech Recognition | Automatic Speech Recognition (ASR), also known as Speech to Text (STT), is the task of transcribing a given audio to text. It has many applications, such as voice user interfaces. | ['audio'] |
| Audio-to-Audio | Audio-to-Audio is a family of tasks in which the input is an audio and the output is one or multiple generated audios. Some example tasks are speech enhancement and source separation. | ['audio'] |
| Audio Classification | Audio classification is the task of assigning a label or class to a given audio. It can be used for recognizing which command a user is giving or the emotion of a statement, as well as identifying a speaker. | ['audio'] |
| Image Editing | Image editing is the task of modifying an image to match a given text description. It can be used to modify the attributes of an image, such as the color of an object or the background. | ['text', 'image'] |

Table 17: Multimedia tools and their descriptions

| Name | Description | Parameters |
|---|---|---|
| Image Downloader | Downloads an image from a given URL. | ['url'] |
| Video Downloader | Downloads a video from a given URL. | ['url'] |
| Audio Downloader | Downloads an audio file from a given URL. | ['url'] |
| Text Downloader | Downloads the text content from a given URL. | ['url'] |
| Text Search | Searches for a specific text or keyword on the internet. | ['text'] |
| Image Search | Searches for images on the internet based on a given query. | ['text'] |
| URL Extractor | Extracts URL from text | ['text'] |
| Video Search | Searches for videos on the internet based on a given query. | ['text'] |
| Text-to-Video | Generates a video based on a given text description. | ['text'] |
| Text-to-Audio | Generates an audio file based on a given text description. | ['text'] |
| Image-to-Text | Extracts text from an input image using Optical Character Recognition (OCR). | ['image'] |
| Audio-to-Text | Transcribes speech from an audio file into text. | ['audio'] |
| Video-to-Text | Transcribes speech from a video file into text. | ['video'] |
| Audio Noise Reduction | Reduces background noise or unwanted sounds from a given audio file. | ['audio'] |
| Audio Effects | Applies various audio effects to a given audio file according to human instruction, such as reverb, chorus, or equalization. | ['audio', 'text'] |
| Audio Splicer | Combines two audio files into a single output file. | ['audio', 'audio'] |
| Voice Changer | Modifies the characteristics of a recorded voice according to human instruction, such as tone, pitch, or gender. | ['audio', 'text'] |
| Text Summarizer | Summarizes a given text into a shorter version while retaining the main points. | ['text'] |
| Text Translator | Translates a given text from one language to english. | ['text'] |
| Text Sentiment Analysis | Analyzes the sentiment of a given text, identifying if it is positive, negative, or neutral. | ['text'] |
| Text Grammar Checker | Checks a given text for grammatical errors and suggests corrections. | ['text'] |
| Text Simplifier | Rewrites a given text in a simpler and more understandable manner. | ['text'] |
| Keyword Extractor | Extracts the most important keywords and phrases from a given text. | ['text'] |
| Text Paraphraser | Rewrites a given text using different words while maintaining its original meaning. | ['text'] |
| Topic Generator | Generates a list of relevant topics or ideas based on a given input. | ['text'] |
| Audio-to-Image | Generates an image that visually represents a given audio, such as a waveform or spectrogram. | ['audio'] |
| Video-to-Audio | Extracts the audio track from a given video file. | ['video'] |
| Video-to-Image | Extracts a still image from a given video. | ['video'] |
| Image Stitcher | Stitches together two input images to create a panorama or collage. | ['image', 'image'] |
| Image Colorizer | Adds color to a black and white input image using deep learning techniques. | ['image'] |
| Video Stabilizer | Stabilizes a shaky input video to produce a smoother output video. | ['video'] |
| Video Speed Changer | Adjusts the playback speed of a given video according to human instruction, either speeding it up or slowing it down. | ['video', 'text'] |
| Video Synchronization | Synchronizes the timing of an existing voiceover or audio file with the visuals of a given video. | ['video', 'audio'] |

Table 18: Daily Life APIs and their descriptions

| API Name | API Description | Parameter Names |
|---|---|---|
| get_news_for_topic | Get the news for a specific topic | ['topic'] |
| stock_operation | Do a specific operation on a specific stock | ['stock', 'operation'] |
| book_flight | Book a flight for a specific date, from a specific location to a specific destination | ['date', 'from', 'to'] |
| book_hotel | Book a specific hotel for a specific date | ['date', 'name'] |
| book_car | Book a car for a specific date, in a specific location | ['date', 'location'] |
| online_shopping | Buy a product from a specific website | ['website', 'product'] |
| send_email | Send an email to a specific email address | ['email_address', 'content'] |
| send_sms | Send an sms to a specific phone number | ['phone_number', 'content'] |
| share_by_social_network | Share a specific content by a specific social network | ['content', 'social_network'] |
| book_restaurant | Book a specific restaurant for a specific date | ['date', 'name'] |
| search_by_engine | Search a specific query by a specific search engine | ['query', 'engine'] |
| apply_for_job | Apply for a specific job | ['job'] |
| see_doctor_online | See a specific doctor for a specific disease | ['disease', 'doctor'] |
| consult_lawyer_online | Consult a specific lawyer for a specific legal issue | ['issue', 'lawyer'] |
| enroll_in_course | Enroll in a specific course at a specific university | ['course', 'university'] |
| buy_insurance | Buy a specific insurance from a specific insurance company | ['insurance', 'company'] |
| online_banking | Do a specific banking operation online at a specific bank | ['instruction', 'bank'] |
| daily_bill_payment | Pay a specific bill | ['bill'] |
| sell_item_online | Sell a specific item at a specific online store | ['item', 'store'] |
| do_tax_return | Do the tax return for a specific year | ['year'] |
| apply_for_passport | Apply for a passport | ['country'] |
| pay_for_credit_card | Pay for a specific credit card | ['credit_card'] |
| auto_housework_by_robot | Let a robot do a housework by following a specific instruction | ['instruction'] |
| auto_driving_to_destination | Let a car drive to a specific destination | ['destination'] |
| deliver_package | Deliver a specific package to a specific destination | ['package', 'destination'] |
| order_food_delivery | Order a specific food to be delivered to a specific location at a specific platform | ['food', 'location', 'platform'] |
| order_taxi | Order a taxi to a specific location at a specific platform | ['location', 'platform'] |
| play_music_by_title | Play a specific music by a specific title | ['title'] |

