# OpenReview forum: "TaskBench: Benchmarking Large Language Models for Task Automation"
_ICLR.cc/2024/Conference — Submitted to ICLR 2024_

### Official Review · Reviewer_CSek · 2023-10-13

**Soundness:** 1 poor
**Presentation:** 3 good
**Contribution:** 2 fair
**Rating:** 3
**Confidence:** 4

**Summary:**

This work introduces a benchmark called TaskBench to evaluate LLMs for task automation. It includes three stages: task decomposition, tool invocation, and parameter prediction. In particular, it considers tools in a graph structure, which could motivate more complicated applications of LLM for task automation. It leverages back-instruct to automatically create the test cases. Extensive evaluation show the utility of this benchmark in evaluting LLMs' capability of task automation.

**Strengths:**

1. The introduction of an open-source benchmark with tool graph is nice, it fills the blank in the field. And the three scenarios included are interesting and realistic
2. The back-instruct technique and the idea of sampling a subgraph from the whole task graph to build test cases are intuitive and sound
3. The experiments cover a wide range of both black-box and open-source models with various metrics

**Weaknesses:**

1. I like the back-instruct, but LLM-generated queries can be biased and not well-aligned with user behavior. In addition, for the benchmark to be faithful (which is very important given that benchmark is what people would use to assess model in a relatively long period), the query and answer should be matched, I doubt whether the rule-based and LLM-based check could always ensure the correctness of the generated instruction. I do think human verification on all the generated instruction can make the benchmark more faithful.

**I took a close look at the huggingface dataset produced in this paper, a large portion of them are incorrect. For example, one of the data (id: 10949228), the user query  is "I have an image 'example.jpg' containing some information. I want to convert the image content into text and then answer the question: 'What is the main topic of the image?", the tool graph is Image-to-text -> VQA. However, once you do image-to-text (assuming captioning), why VQA? it makes no sense. it can just be a textual QA, right? There are many similar cases in the dataset.**

 2. I don't quite understand the goal of evaluating task decomposition as textual descriptions. I do think evaluating task decomposition is important, but as long as it can build the correct tool graph and predict the correct parameter, it is good enough. So I don't think evaluating textual description is necessary, and given how diverse the textual description could be, the current automatic evaluation metric could be misleading.

**Questions:**

See weakness

1. when you build a DAG tool graph, do you consider one graph that contains multiple disconnected DAG?
2. Are tool graphs with more nodes more challenging? I think analysis of the correlation between the difficulty of the test cases and their numbers of nodes is interesting.

---

> ### Author Response · Authors · 2023-11-20
> **Response - 0**
>
> Thanks for your insightful comments. Below are our responses to your concerns:
>
> ### **[W1] Human verification on Back-Instruct Dataset**
>
> ### **Response:**
>
> Regarding the benchmark's faithfulness and the correctness of generated instructions, we acknowledge your concern. Compared with previous methods, we further employ rule-based and LLM-based checks to filter out low-quality samples. Following your suggestions, we are exploring the integration of human verification to enhance accuracy and reliability. This combined approach will ensure a more robust and faithful benchmark. We conducted a human evaluation of the current version of the dataset and put them in Appendix A.3, please refer to [G1] in General Response.
>
> ### **[W2] Error Analysis on Back-Instruct Dataset**
>
> ### **Response:**
>
> Thank you for pointing this out. We acknowledge the error in the specific instance you highlighted (id: 10949228). This example illustrates a mismatch of parameter types between VQA and the image-to-text tool: the output of image-to-text is incorrectly linked to the image input of VQA. We conducted a human verification on the constructed dataset, and the results are shown in [G2] of the general response. In our examination, we found the proportion of error cases is nearly 12%. After human verification, we found that our dataset could provide a better quality. We will continue to improve it in two ways:
>
> - **Improving the quality of tool graphs.** Our analysis shows that these mismatches arise primarily when parameter types are not explicitly defined within the tool graph, leading to incorrect assumptions by the LLM about resource types. To rectify this, we propose introducing resource-type nodes in our tool graph. This will clearly delineate the types of resources each tool handles, ensuring that tool nodes are connected in a way that makes logical sense for the given query. This should significantly reduce such mismatches.
> - **Perform full human verification.** We recognize the value of human verification in ensuring the accuracy and faithfulness of our benchmark. We are exploring ways to integrate this into our process, alongside our automated checks.
>
> ### **[W3] The importance of Evaluating task decomposition as textual descriptions.**
>
> ### **Response:**
>
> Thank you for your comments and concerns regarding the necessity of evaluating task decomposition. We appreciate your perspective and it is still challenging to determine the most appropriate metric to measure task decomposition. Since no one has explored the evaluation of task decomposition when conducting task automation, we want to explore a new perspective to analyze task decomposition by using textual descriptions. Just as shown in Figure 1 in our paper, the textual descriptions from task decomposition will instruct LLMs to parse tools and predict parameters. Therefore, we think that evaluating textual descriptions can also draw some inspiration to enhance LLMs in the future and reflect the planning capability of LLMs to obtain the decomposed sub-tasks. However, we also acknowledge your concerns that current metrics based on textual descriptions are still not enough to fully reflect the accuracy of task decomposition. We will continue to design better metrics to evaluate task decomposition, including subjective and objective evaluations.

---

> > ### Author Response · Authors · 2023-11-20
> > **Response - 1**
> >
> > ### **[Q1] When you build a DAG tool graph, do you consider one graph that contains multiple disconnected DAG?**
> >
> > ### **Response:**
> >
> > Thanks for your question. Yes, when we sample DAG-structured tool subgraphs from the complete tool graph, we also consider scenarios that include multiple disconnected DAGs. Such scenarios represent a category of practical commands, where the potential tool invocation graph comprises several independent sets of tool invocations. In our approach, Back-Instruct is capable of understanding the dependencies within the tool subgraphs and can create separate sub-commands accordingly. For instance, in the example of Table 12, Audio Noise Reduction, Audio Effects, Audio Splicer, Audio-to-Text, and Image-to-Text, Text Sentiment Analysis, Text Search form two independent groups of tool combinations. Correspondingly, Back-Instruct generates instructions for two sub-commands: audio processing and image processing.
> >
> > | sampled tool graph | Instruction | Tool Invocation Graph |
> > | --- | --- | --- |
> > | {"sampling_method": "dag", "number_of_tools": 7, "sampled_nodes": [{"id": "Audio Effects", "input-type": ["audio", "text"], "output-type": ["audio"]}, {"id": "Audio Noise Reduction", "input-type": ["audio"], "output-type": ["audio"]}, {"id": "Audio Splicer", "input-type": ["audio", "audio"], "output-type": ["audio"]}, {"id": "Audio-to-Text", "input-type": ["audio"], "output-type": ["text"]}, {"id": "Image-to-Text", "input-type": ["image"], "output-type": ["text"]}, {"id": "Text Search", "input-type": ["text"], "output-type": ["text"]}, {"id": "Text Sentiment Analysis", "input-type": ["text"], "output-type": ["text"]}], "sampled_links": [{"source": "Audio Noise Reduction", "target": "Audio Effects"}, {"source": "Audio Splicer", "target": "Audio Noise Reduction"}, {"source": "Audio Splicer", "target": "Audio-to-Text"}, {"source": "Image-to-Text", "target": "Text Sentiment Analysis"}, {"source": "Text Sentiment Analysis", "target": "Text Search"}]} | I have an audio file 'example.wav' which has some background noise, please clean it up, transcribe it to text, and analyze its sentiment. Also, I need to extract text from an image 'example.jpg' and search for that text on the internet. Finally, please combine the cleaned audio file with another audio file 'example2.wav' and apply a reverb effect to create the final audio output. | {"nodes": [{"id": "Audio Effects", "input": ["Audio Splicer", "reverb"]}, {"id": "Audio Noise Reduction", "input": ["example.wav"]}, {"id": "Audio Splicer", "input": ["Audio Noise Reduction", "example2.wav"]}, {"id": "Audio-to-Text", "input": ["Audio Noise Reduction"]}, {"id": "Image-to-Text", "input": ["example.jpg"]}, {"id": "Text Search", "input": ["Image-to-Text"]}, {"id": "Text Sentiment Analysis", "input": ["Audio-to-Text"]}], "links": [{"source": "Audio Noise Reduction", "target": "Audio Splicer"}, {"source": "Audio Noise Reduction", "target": "Audio-to-Text"}, {"source": "Audio Splicer", "target": "Audio Effects"}, {"source": "Audio-to-Text", "target": "Text Sentiment Analysis"}, {"source": "Image-to-Text", "target": "Text Search"}]} |
> >
> > *Table 12. Back-Instructed on DAG with Multiple Disconnected DAGs*

---

> > > ### Author Response · Authors · 2023-11-20
> > > **Response - 2**
> > >
> > > ### **[Q2] Are tool graphs with more nodes more challenging? I think the analysis of the correlation between the difficulty of the test cases and their numbers of nodes is interesting.**
> > >
> > > ### **Response:**
> > >
> > > Yes. The difficulty of LLM to perform task automation is positively correlated with the number of tool nodes that the tool graphs contain. To make a clear understanding of the correlation between the number of nodes in the tool graph and the performance of LLMs in task automation, we conduct a detailed analysis in Table 8. This analysis includes various metrics such as node set accuracy, edge set accuracy, and graph accuracy, which measure the exact-match accuracy of the node set, edge set, and the entire graph, respectively.
> > >
> > > | # tool nodes | # supports | node set accuracy  | edge set accuracy  | graph accuracy |
> > > | --- | --- | --- | --- | --- |
> > > | 1 (single) | 2059 | 96.16 | - | 96.16 |
> > > | 2 | 278 | 86.33 | 84.53 | 84.53 |
> > > | 3 | 1313 | 67.93 | 60.70 | 60.39 |
> > > | 4 | 1280 | 64.29 | 75.62 | 54.37 |
> > > | 5 | 731 | 54.03 | 70.53 | 41.58 |
> > > | 6 | 290 | 50.34 | 39.31 | 39.31 |
> > > | 7 | 151 | 49.66 | 36.42 | 36.42 |
> > > | 8 | 60 | 35.00 | 25.00 | 25.00 |
> > > | 9 | 55 | 38.18 | 21.81 | 21.81 |
> > > | 10 | 64 | 39.06 | 31.25 | 31.25 |
> > > | overall | 6281 | 73.52 | 67.55 | 67.25 |
> > >
> > > *Table 13.  Task automation performance with different number of tools on gpt-4*
> > >
> > > From Table 13, we observed that as the number of tools in the tool graphs increases, there is a clear downward trend in various performance metrics such as node set accuracy, edge set accuracy, and graph accuracy. This trend confirms that tool graphs with a higher number of tools present more complex challenges for LLMs in task automation. Specifically, the result shows a significant drop in performance metrics when moving from simpler graphs (1-2 tools) to more complex ones (3 or more tools). For instance, while single-node graphs achieve a high graph accuracy of 96.16%, this metric falls to 39.31% for graphs with 6 tools and further decreases to 25.00% for 8-node graphs. This correlation between the number of tools and the difficulty of the test cases can be attributed to the increased complexity in understanding and processing more extensive and intricate links between tools. As the number of tools grows, LLMs must handle a larger set of possible dependencies, which significantly challenges their predictive and analytical capabilities. The results from this analysis underline the importance of continuous advancements in LLM capabilities to be consistent with the increasing complexity of tasks in various domains.

---

> ### Comment · Reviewer_CSek · 2023-11-20
>
> I appreciate your response. Is the dataset in the supplementary materials updated? If so, I would do another pass and will consider raising the score based on the updated dataset.

---

> > ### Author Response · Authors · 2023-11-22
> > **Response**
> >
> > Thanks for your constructive comments, which have significantly improved the quality of our datasets. We have completed human verification of the datasets and already updated them in the folder  `datasets_human_verified` in our supplementary materials. Your insights are invaluable to us, and we are eager to address any remaining issues.

---

> > > ### Comment · Reviewer_CSek · 2023-11-22
> > >
> > > Thanks! I checked the updated dataset very quickly. There are still many errors, and even though they may only take up around 10% of the dataset, it is still unacceptable for a benchmark. (These errors can be detected simply by rule-checking.)
> > >
> > > For example, many arguments are wrong for the huggingface dataset:
> > >
> > > Image-to-Image requires 1 args only 2 are provided
> > > Required: ['image']
> > > Predicted: ['example.jpg', 'target.png']
> > >
> > > Text-to-Video requires 1 args only 2 are provided
> > > Required: ['text']
> > > Predicted: ['The life cycle of a butterfly is truly fascinating. From a tiny egg to a beautiful butterfly, the journey is full of incredible transformations.', 'example.jpg']
> > >
> > > Text-to-Video requires 1 args only 2 are provided
> > > Required: ['text']
> > > Predicted: ['<node-3>', 'example.mp4']
> > >
> > > Translation requires 1 args only 2 are provided
> > > Required: ['text']
> > > Predicted: ['<node-0>', '<user_question>']
> > >
> > > Document Question Answering requires 2 args only 1 are provided
> > > Required: ['image', 'text']
> > > Predicted: ['<node-2>']

---

> ### Author Response · Authors · 2023-11-23
> **Response**
>
> Thanks for your suggestions. We have further designed rules to filter samples based on the mismatch in the parameter number you pointed out：
>
> - On the DailyLife APIs domain, the ratio of error to total samples is: 30 / 4350 = 0.7 %.
> - On the Hugging Face Tools domain, the ratio of error to total samples is: 558 / 8104 = 6.8%.
> - On the Multimedia Tools domain, the ratio of error to total samples is: 303 / 5887 = 5.1%.
>
> We have updated the dataset in the supplementary material. We appreciate the thorough examination you provided, which has significantly enhanced the quality of our dataset. Your insights are invaluable to us, and we are committed to addressing any remaining issues.

---

> ### Author Response · Authors · 2023-11-23
> **Follow-Up: Seeking Further Feedback**
>
> Thanks for your continuously constructive comments on our paper. We have updated our dataset by further refining our datasets with rule-based and human verification. Since the deadline of the rebuttal is approaching, we are looking forward to your feedback. We are also willing to answer all of your concerns if possible.
>
> Thank you again for your time and effort in reviewing.

---

### Official Review · Reviewer_peHw · 2023-10-31

**Soundness:** 2 fair
**Presentation:** 3 good
**Contribution:** 3 good
**Rating:** 5
**Confidence:** 4

**Summary:**

The paper proposes a new benchmark for evaluating large language models' capabilities in completing user requests by utilizing external tools. The benchmark is constructed by first representing a collection of tools as a graph, sampling subgraphs representing valid tasks, and finally back-instructing GPT-4 in generating corresponding user requests based on the sampled subgraphs. The evaluation shows that current LLMs can still struggle in predicting correct tool-use plans for complex tasks.

**Strengths:**

- The paper introduces a timely benchmark for evaluating LLMs' tool-use capabilities. It's a good effort to make the evaluation more standardized since there is an increasing volume of work in this area.
- Representing tools as graphs is interesting, since the graph structure allows more diverse and complex tasks. The authors also include 3 sets of tools for different scenarios.
- Back-instruct is a natural approach to construct the benchmark. However, I have concerns on the resulting data quality (see below).

**Weaknesses:**

- One of my concern is that depending on the subgraph sampling procedure (which is not clearly described in the paper), the task might be unnatural (deviating from what users would ask in real-world) while being valid. For example, one can always make a complex task by combining many tools "Tool 1" --> "Tool 2" --> ...."Tool N", while the task is rarely encountered in real-world. Does the sampling procedure take "task naturalness" into account? Also, I'd suggest to include some actual examples in the paper.
- While back-instructing to generate user requests is an intuitive approach. This procedure relies on GPT-4 and there is no guarantee that the generated data is correct (e.g., many of the examples are filtered out by simply using rule-based critics as mentioned). Even if more advanced LLM-based critic is used, there may still be wrong examples. Since the dataset is intended for evaluation, the data quality is of most the utmost importance. Without further (potentially manual) verification, it is not very convincing on the reliability of the benchmark.
- As discussed in related work, there are several parallel efforts in creating benchmarks for LLM tool usage, and perhaps the most related is ToolBench [1]. Can the authors provide more detailed discussion/comparison to the work, and highlight the contribution in this work?

[1] Tool learning with foundation models. Qin et al. 2023.

**Questions:**

- Tool graph construction: is there an example on the constructed tool graph before subsampling?
- In Figure 3, GPT-4 almost reaches 100% in performance, would the benchmark will be saturated soon, perhaps when the next generation LLMs come out?

---

> ### Author Response · Authors · 2023-11-20
> **Response**
>
> Thanks for your insightful comments. Below are our responses to your concerns:
>
> ### **[W1] "task naturalness" in sampling procedure? And add more actual examples in the paper.**
>
> ### **Response:**
>
> Thanks for your question. Here we mainly consider "task naturalness" in the instruction generation stage since currently, it is challenging to consider "task naturalness" in sampling without any human supervision. Therefore, to improve the naturalness, we adopt LLM-based and rule-based critics to filter out instances that rarely occur in the real world. We will follow your suggestions to introduce "task naturalness" into the sampling stage by using LLM critics. Besides, following your and other reviewers' suggestions, we have conducted human verification on our datasets to detect errors (Please see [G1] & [G2] of General Response) and further improve the quality of our datasets. Besides, we have presented concrete samples in [G1] of General Response and put them in Appendix A.4.1 of the revised version.
>
> ### **[W2] Further (manual) verification to verify the reliability of the benchmark.**
>
> ### **Response:**
>
> Thanks for your suggestions. We admit it will be better with further human verification. We have conducted human verification and analyzed error patterns in our datasets (Please see [G1] and [G2] of General Response). Besides, compared with previous datasets (e.g., ToolBench) which all used instruction methods, we further conducted LLM-based and rule-based critics (Table 1) to filter out low-quality samples and reported the agreement of LLM rankings under human and TaskBench (Table 7). Due to the limited rebuttal period, we will continue to refine the reliability of our benchmark by using subjective and objective evaluations.
>
> ### **[W3] More discussions or comparisons to the concurrent work (ToolBench).**
>
> ### **Response:**
>
> Thanks for the suggestion. Due to the page limitation, we have put some discussions between the concurrent work in the appendix in the initial version. To better understand the differences between TaskBench and other concurrent work, we have conducted in-depth comparisons in Appendix A.2 of our latest version from the perspective of datasets and evaluations. Please see [G3] in General Response to the differences and the advantages of our method.
>
> ### **[Q1] Is there an example of the constructed tool graph before subsampling?**
>
> ### **Response:**
>
> Thanks for your question. In Appendix A.10 of the revised version, we have added a visualization of the constructed tool graph on Multimedia domain for reference, where the nodes denote tools, and directed edges indicate that the output type of the source tool matches the input type of the target tool.
>
> ### **[Q2] In Figure 3, GPT-4 almost reaches 100% in performance, Will the benchmark be saturated soon, perhaps when the next generation of LLMs comes out?**
>
> ### **Response:**
>
> Thanks for your question. We guess this is because our evaluation dataset is also generated by GPT-4, so that its performance will be overestimated when also using GPT-4 for testing. Therefore, we also involve human evaluation to validate its quality. Besides, we believe that when more powerful LLMs come out, the dataset generated by these LLMs will be better and demonstrate better alignments to human preference when compared with GPT-4.

---

> > ### Comment · Reviewer_peHw · 2023-11-20
> > **Thank you for the response**
> >
> > Thank you to the authors for the response. I appreciate the authors' detailed response and efforts in further verification on the data quality. As data quality is my main concern and the manual verification is still ongoing, I would remain my original score.

---

> > > ### Author Response · Authors · 2023-11-22
> > > **Response**
> > >
> > > Thanks for your constructive comments, which have significantly improved the quality of our datasets. We have completed human verification of the datasets and already updated them in the folder  `datasets_human_verified` in our supplementary materials. Your insights are invaluable to us, and we are eager to address any remaining issues.

---

> ### Author Response · Authors · 2023-11-23
> **Follow-Up: Seeking Further Feedback**
>
> Thanks for your constructive suggestions on our paper. Following your suggestions and other reviewers’ comments, we have updated our dataset by further refining our datasets with rule-based and human verification. Since the deadline of the rebuttal is approaching, we are looking forward to your feedbacks and examinations on TaskBench. We are also willing to answer all your concerns if possible.
>
> Thank you again for your time and effort in reviewing.

---

### Official Review · Reviewer_BEYM · 2023-11-01

**Soundness:** 3 good
**Presentation:** 3 good
**Contribution:** 2 fair
**Rating:** 6
**Confidence:** 4

**Summary:**

The paper generates a tool usage dataset for evaluating LLM-based autonomous agents. They  introduce Tool Graph to represent the decomposed tasks, and adopt a back-instruct method to generate instructions. The evaluation is conducted from task decomposition, tool invocation, and parameter prediction.

**Strengths:**

+ The paper is well-written and easy to read.

+ The paper studies an important question, which very interesting under the era of autonomous AI agent.

**Weaknesses:**

- The paper focuses on the evaluation of task solving (agent) capabilities of different LLMs. Although the experiments are enough, the analyses are partial. For instance, which contributes to the performance of different LLMs? What are the intrinsic difference between different existing LLMs in performing agent tasks? Which findings can we derive to better improve the capabilities of current open-source LLMs? Personally,  insights from the evaluation results are somehow shallow.

- For an evaluation paper, I think more diverse LLMs should be evaluated as well, such as Claude-2. I also expect a case study to show the performance gap among different LLMs.

- Missing discussion with a very relevant work ToolLLM [1]. The workflow is quite similar: preparing high quality tools (or APIs), back-generate the instructions that involve these APIs, and annotate how LLMs solve these instructions. The main difference is that this paper involves a concept of Tool Graph when organizing the structure of tools/APIs. Though ToolLLM can be considered as a concurrent work,  I think the authors should discuss their core differences and the unique advantage of this submission (experiments, findings, etc.).

[1] Qin Y, Liang S, Ye Y, et al. Toolllm: Facilitating large language models to master 16000+ real-world apis[J]. arXiv preprint arXiv:2307.16789, 2023.

**Questions:**

Please response to the weaknesses and make more analysis to the experimental results.

---

> ### Author Response · Authors · 2023-11-20
> **Response - 0**
>
> Thanks for your insightful comments. Below are our responses to your concerns:
>
> ### **[W1] More Analyses of evaluation results.**
>
> ### **Response:**
>
> Thank you for your feedback. We acknowledge your concerns regarding the perceived partiality in our analyses. To resolve your concerns, we have conducted further analyses of the aspects you highlighted:
>
> **1) Factors Contributing to Task Automation Performance:**
>
> - **Reasoning**: The capacity for complex problem-solving and reasoning varies significantly among LLMs. This ability is crucial for understanding and executing complex tasks. In mathematical and coding tasks, GPT series models show stronger reasoning ability, therefore they also possess corresponding capabilities in task planning and tool utilization.
> - **Instruction Following**: The proficiency in comprehending and adhering to user instructions differs, influencing task execution efficiency. Our analysis reveals that models fine-tuned with instruction-following settings, such as Vicuna-13b, WizardLLM-13b, and Nous-Hermes-13b, outperform the baseline Llama-2-13b model in task automation. Specifically,  we observe that WizardLLM-13b can perform better than Vicuna-13b since it fine-tunes more complex instructions. These results also demonstrate the necessity of instruction following.
>
> **2) Intrinsic Differences of LLMs in Performing Agent Tasks:**
>
> - **Code Pre-training:** We note that the models (Code-Llama) with more code-pretraining outperform other open-source LLMs in task automation. Experimental results show an average improvement of 4.45% in tool prediction (\textit{n-f1}) and 12.76% in parameter prediction (\textit{v-f1}) across various domain datasets. We conclude that since task automation usually involves multiple stages and tools, using code-style / structural text as the interface to switch different stages is better.
> - **Alignment Techniques**: Besides, the models (e.g., GPT series models) with human alignments (e.g., RLHF) demonstrate stronger task automation capabilities than open-source LLMs. These results also indicate that RLHF allows large language models to develop more generalized reasoning abilities, reducing overfitting to specific instructions.
>
> Based on our analysis, we propose the following improvements for open-source LLMs:
>
> - **Enhanced Code Pre-training**: Strengthening code pre-training can significantly boost the performance of LLMs in task automation.
> - **Quality of Instructions**: Developing high-quality (complex and diverse) instructions is crucial for crafting more efficient LLMs.
> - **Ongoing Knowledge Acquisition**: Encouraging continuous learning and knowledge updating can make LLMs more adaptable and proficient.
>
> We have incorporated these additional analyses in Section 3.5 of the revised version to offer a more comprehensive view of the capabilities and development of LLMs in task automation. This should provide a deeper understanding and address the concerns you have raised about the initial shallowness of our insights.

---

> ### Author Response · Authors · 2023-11-20
> **Response - 1**
>
> ### **[W2] Evaluating a Broader Range of LLMs, Including Claude-2.**
>
> ### **Response:**
>
> Thank you for your valuable suggestion. In response, we have incorporated the evaluation results of Claude-2 into our analysis. From Table 5-9, we observed that Claude-2 exhibited superior performance compared to gpt-3.5-turbo, yet it did not match the capabilities of GPT-4. Due to the limited time of the rebuttal period, we will add more diverse LLMs to our paper in the final version.
>
> |  | R1↑  | R2↑  | BsF↑ |
> | --- | --- | --- | --- |
> | Multimedia | 49.19 | 23.71 | 89.22 |
> | Hugging Face | 44.49 | 21.37 | 88.71 |
> | Daily APIs | 81.28 | 68.06 | 95.65 |
>
> *Table 5. Task Decomposition Performance of Claude-2.*
>
> |  | n-f1↑ | t-f1↑ | v-f1↑ |
> | --- | --- | --- | --- |
> | Multimedia | 66.15 | 53.57 | 23.79 |
> | Hugging Face | 69.77 | 48.29 | 32.33 |
> | Daily APIs | 79.73 | 78.14 | 60.10 |
>
> *Table 6. Tool Invocation and Parameter Prediction Performance of Claude-2 in node-structure Tasks.*
>
> |  | n-f1↑ | e-f1↑ | t-f1↑ | v-f1↑ |
> | --- | --- | --- | --- | --- |
> | Multimedia | 82.69 | 60.76 | 75.83 | 59.59 |
> | Hugging Face | 80.07 | 48.61 | 67.10 | 45.81 |
> | Daily APIs | 95.62 | 62.17 | 95.13 | 64.95 |
>
> *Table 7. Tool Invocation and Parameter Prediction Performance of Claude-2 in chain-structure Tasks.*
>
> |  | n-f1↑ | e-f1↑ | t-f1↑ | v-f1↑ |
> | --- | --- | --- | --- | --- |
> | Multimedia | 85.47 | 63.42 | 77.70 | 60.61 |
> | Hugging Face | 83.17 | 52.66 | 69.91 | 47.93 |
> | Daily APIs | 93.76 | 55.96 | 92.49 | 64.69 |
>
> *Table 8. Tool Invocation and Parameter Prediction Performance of Claude-2 in DAG-structure Tasks.*
>
> |  | n-f1↑ | e-f1↑ | t-f1↑ | v-f1↑ |
> | --- | --- | --- | --- | --- |
> | Multimedia | 81.07 | 56.62 | 73.08 | 54.40 |
> | Hugging Face | 79.15 | 45.19 | 64.52 | 43.87 |
> | Daily APIs | 93.73 | 57.66 | 92.89 | 64.47 |
>
> *Table 9. Tool Invocation and Parameter Prediction Performance of Claude-2 in All Tasks.*
>
> ### **[W3] A case study to show the performance gap among different LLMs.**
>
> ### **Response:**
>
> Thanks for your suggestion. We added a case study to show the performance difference between different LLMs, please refer to General Response [G2] and Appendix A.4.2 in the revised version.
>
> ### **[W4] Discussion with ToolLLM.**
>
> ### **Response:**
>
> Thanks for the suggestion. Due to the page limitation, we have put some discussions between the concurrent work in the appendix in the initial version. To better understand the differences between TaskBench and other concurrent work, we have conducted in-depth comparisons in Appendix A.2 of the latest version from the perspective of datasets and evaluations. Please see [G3] in General Response to the differences and the advantages of our method.

---

> > ### Comment · Reviewer_BEYM · 2023-11-21
> >
> > Thanks for providing the additional experiments and analyses. But considering the data quality issues raised by two reviewers, I will maintain my score.

---

> > > ### Author Response · Authors · 2023-11-22
> > > **Response**
> > >
> > > Thanks for your constructive comments, which have significantly improved the quality of our datasets. We have completed human verification of the datasets and already updated them in the folder  `datasets_human_verified` in our supplementary materials. Your insights are invaluable to us, and we are eager to address any remaining issues.

---

> > > ### Author Response · Authors · 2023-11-23
> > > **Follow-Up: Seeking Further Feedback**
> > >
> > > Thanks for your constructive suggestions on our paper. Following your suggestions and other reviewers’ comments, we have updated our datasets by further refining our datasets with rule-based and human verification. Since the deadline of the rebuttal is approaching, we are looking forward to your feedbacks and examinations on TaskBench. We are also willing to answer all your concerns if possible.
> > >
> > > Thank you again for your time and effort in reviewing.

---

### Official Review · Reviewer_LVvj · 2023-11-01

**Soundness:** 3 good
**Presentation:** 3 good
**Contribution:** 3 good
**Rating:** 5
**Confidence:** 4

**Summary:**

This work focuses on creating a benchmark that evaluates the task automation for large language models. A common practice is to formulate it into three critical stages, including task decomposition, tool invocation and parameter prediction.
The main contributions of the work are (1) The dataset creation, denoted as TaskBench, based on the aforementioned formulation. (2) Based on the created dataset, the performance of different aspects can be evaluated effectively and quantitatively.
For dataset creation, in details, to facilitate the dataset construction, the authors introduced the concept of Tool Graph to represent the connections/dependencies among the decomposed tasks. Three resources are leveraged for collecting tools, including the HuggingFace (e.g. Summarization), Multimedia (e.g. Text-to-Video), and Daily Life APIs (e.g. stock operation).
With pre-defined tools, the authors formulate three patterns for tool invocation: Node, Chain, and DAG (directed acyclic graph). With the diverse sampled subgraphs, back-instruct method is used to inversely craft user instructions, task steps, and tool invocation graphs.
For task evaluation, different steps are evaluated. Rouge-* and bertScore are used for evaluating textual description in task decomposition. F1 is used for evaluating the tool invocation and tool parameter prediction.
Experimental results demonstrate the TaskBench can be effectively utilized to evaluate task automation ability of LLMs.

**Strengths:**

This work covers more diverse tools than some other related work. For example, ToolQA defined 13 tools, mainly for accessing external knowledge. In this work, the authors consider three tool resources including the huggingface, multimedia and daily life APIs, in total 103 tools. Furthermore, designing LLM-based critic and rule-based critic is great to evaluate the consistency of the generated tool invocation graphs with the sampled tool subgraphs, without too much human effort.
The experimental results are also interesting. In terms of zero-shot, the OpenAI model significantly outperforming the open-sourced LLM. But for few-shot setting, code-llama gets closer to the OpenAI models. To my understanding, this may indicate that the OpenAI models did pretty good SFT and RLHF, to make models understand the instructions/task better, which aligns with the finding from “In-Context Learning Creates Task Vectors”.

**Weaknesses:**

Although there is a section discussing the positive correlation of the proposed evaluation with human assessment and a section about using LLM and rule to check the alignment between the generated data and sampled tool sub-graph, in terms of data quality, it would be more insightful to show the quality measured by human. This will show the quality of the self-critic, either LLM-based critic or rule-based critic.
If the authors can provide more examples (predictions and gold answers) in appendix, it would be useful to help readers understand the difficulty of each tasks and some error cases/analysis in main text would be useful. Otherwise, the number itself cannot provide too much information regarding this dataset.

**Questions:**

Comparing to the node prediction, the edge prediction is harder based on the evaluation results. What’s the error types for edge prediction? Does the model make equivalent edge prediction but different connectivity?

---

> ### Author Response · Authors · 2023-11-20
> **Response**
>
> Thanks for your insightful comments. Below are our responses to your concerns:
>
> ### **[W1] In terms of data quality, it would be more insightful to show the quality measured by humans.**
>
> ### **Response:**
>
> Thank you for your suggestion. In the latest version, we have conducted human evaluations on TaskBench to demonstrate its quality in terms of several aspects, including the Naturalness and Complexity of the instructions, and Alignment of the tool invocation graphs. Please see Appendix A.3. The results show that our Back-Instruct strategy produces high-quality user instructions with annotated and executable tool graphs. This approach generates data that is better aligned with human preferences. For more details about the evaluation results, please refer to [G1] in General Response.
>
> ### **[W2] More examples (prediction and gold answers) to understand the difficulty of each task and error cases.**
>
> ### **Response:**
>
> Thanks for your suggestions. To resolve your concerns, we have added more examples in Appendix A.4 to illustrate the difficulty and error cases in conducting task automation based on TaskBench. Please also see our response in [G2] for case studies and error analysis.
>
> ### **[Q1] Compared to the node prediction, the edge prediction is harder based on the evaluation results. What are the error types for edge prediction?**
>
> ### **Response:**
>
> Thanks for your question. Yes, edge prediction is more challenging compared to node prediction because it requires accurate prediction of both the source and target nodes of an edge. Currently, we summarize the main error types in edge prediction as two parts:
>
> - Mismatch of resource types between source node output and target node input.
> - Missing edge prediction between nodes where a resource or temporal dependency exists.
>
> We analyzed 100 randomly sampled examples of edge prediction failures and statistically 27% and 38% of edge prediction errors belong to the above two types respectively.
>
> ### **[Q2] Does the model make equivalent edge prediction but different connectivity?**
>
> ### **Response:**
>
> Generally, if the model has made an equivalent edge prediction, the nodes of this edge will always keep the same connectivity. Besides, in our dataset, such a scenario is rare since we do not involve any repeated calls to a specific tool, leading to unique edge predictions at distinct tool nodes.

---

### Author Response · Authors · 2023-11-20
**General Response - 0**

Dear reviewers,

We sincerely thank each reviewer for providing their constructive comments, which are very helpful in improving our paper. Below are the common concerns of the reviewers and our responses to them:

### **[G1] Need more human evaluation and case studies on the dataset**

### **Response:**

Thanks for the constructive comments of each reviewer. Following your suggestions, we have conducted in-depth human evaluations based on our generated samples and put them in Appendix A.3 of the revised version. To better assess the quality of the datasets constructed by back-instruct, we design three metrics in our evaluation criteria, where two are used to measure the quality of instructions and one is used to evaluate tool invocation graphs:

- **Instructions**
    - **Naturalness:** This metric measures the reasonableness of the instructions, including the commonality of dependencies between tools and their alignment with real-world needs.
    - **Complexity:** This metric assesses the complexity of the instructions, considering factors such as task depth, the number of involved tools, and the relationships between these tools.
- **Tool Invocation Graphs**
    - **Alignment:** Building upon the Feasibility metric, this measures how well the tool invocation graphs align with the instructions, i.e., whether the tool invocation graphs can effectively address the user's commands.

Each metric ranges from 1 to 5, and we design these metrics to assess the effectiveness and faithfulness of our TaskBench in task automation.

To make a fair comparison, we choose two addition baselines to compare our Back-instruct:

- **Back-Instruct (Ours):** we sample tool subgraphs and then backtranslate to instructions and further refine the tool invocation graph.
- **Back-Instruct w/o edges:** compared with our back-instruct, we eliminated edges from our sampled tool subgraphs, preserving only the tool node information in the prompt.
- **Self-Instruct:** based on manually labeled demonstrations and all tools with descriptions, we directly employed GPT-4 to autonomously select tools and then generate the instructions with tool invocation graphs.

During the human evaluation, we randomly selected 50 samples from our TaskBench and invited three domain experts to assess the quality of these samples. To ensure a fair and unbiased evaluation, all samples will be anonymized. We provide canonical samples for these experts to calibrate their criteria during the annotations, and calculate an average score of all experts’ ratings as the final results. All results can be found in Table 1. We observed that all methods (self-instruct or back-instruct) can guarantee the alignment. However, our method, Back-Instruct, scored highest in Naturalness and Complexity. We attribute these superiorities to the realistic resource or temporal dependencies in the sampled tool subgraphs, which allow us to generate more natural instructions in complex scenarios (e.g., multi-tools utilization).

| Methods | Naturalness↑ | Complexity↑ | Alignment↑ | Overall↑ |
| --- | --- | --- | --- | --- |
| Back-Instruct | 3.89 | 4.01 | 3.66 | 3.85 |
| Back-Instruct w/o edges | 3.44 | 3.27 | 3.62 | 3.44 |
| Self-Instruct | 2.18 | 2.01 | 3.64 | 2.61 |

*Table 1. Human evaluation (rating from 1 to 5) on samples constructed by different methods. Average score rating from three human experts.*

---

> ### Author Response · Authors · 2023-11-20
> **General Response - 1**
>
> Moreover, we further draw some cases to intuitively show the differences between the three methods, as shown in Table 2. From these examples, we observe that our Back-Instruct can generate examples with more comprehensive and interconnected tool usage, reflecting higher naturalness and complexity in instruction generation.
>
> | Method | Tools | Instruction | Tool Invocation Graph |
> | --- | --- | --- | --- |
> | Back-Instruct | "sampled_nodes": [{"id": "Audio-to-Image", "input-type": ["audio"], "output-type": ["image"]}, {"id": "Image Colorizer", "input-type": ["image"], "output-type": ["image"]}, {"id": "Image Stitcher", "input-type": ["image", "image"], "output-type": ["image"]}, {"id": "Image Style Transfer", "input-type": ["image", "image"], "output-type": ["image"]}, {"id": "Image-to-Text", "input-type": ["image"], "output-type": ["text"]}], "sampled_links": [{"source": "Image Style Transfer", "target": "Image Stitcher"}, {"source": "Audio-to-Image", "target": "Image Colorizer"}, {"source": "Image Colorizer", "target": "Image Style Transfer"}] | I have an image (example.jpg) with some text on it and an audio file (example.wav). I'd like you to extract the text from the image, generate a visual representation of the audio file, colorize this visual representation using the colors from the original image, apply the visual style of the colorized image to the visual representation of the audio, and finally, stitch these two images together to create a panorama or collage. | {"nodes": [{"id": "Audio-to-Image", "input": ["example.wav"]}, {"id": "Image Colorizer", "input": ["Audio-to-Image"]}, {"id": "Image Stitcher", "input": ["example.jpg", "Image Style Transfer"]}, {"id": "Image Style Transfer", "input": ["Image Colorizer", "Audio-to-Image"]}, {"id": "Image-to-Text", "input": ["example.jpg"]}], "links": [{"source": "Audio-to-Image", "target": "Image Colorizer"}, {"source": "Image Colorizer", "target": "Image Style Transfer"}, {"source": "Image Style Transfer", "target": "Image Stitcher"}]} |
> | Back-Instruct w/o edges | "sampled_nodes": [{"id": "Audio-to-Image", "input-type": ["audio"], "output-type": ["image"]}, {"id": "Image Colorizer", "input-type": ["image"], "output-type": ["image"]}, {"id": "Image Stitcher", "input-type": ["image", "image"], "output-type": ["image"]}, {"id": "Image Style Transfer", "input-type": ["image", "image"], "output-type": ["image"]}, {"id": "Image-to-Text", "input-type": ["image"], "output-type": ["text"]}],  | I would like to create a stylized collage from 'example.jpg' and the sound of the 'example.wav' file. The visual style should be inspired by the similar image search result. The collage should be colorized, and I would like any text in the final image to be extracted. | {"nodes": [{"id": "Audio-to-Image", "input": ["example.wav"]}, {"id": "Image Colorizer", "input": ["Image Style Transfer"]}, {"id": "Image Search (by Image)", "input": ["example.jpg"]}, {"id": "Image Stitcher", "input": ["Audio-to-Image", "example.jpg"]}, {"id": "Image Style Transfer", "input": ["Image Stitcher", "Image Search (by Image)"]}, {"id": "Image-to-Text", "input": ["Image Colorizer"]}], "links": [{"source": "Audio-to-Image", "target": "Image Stitcher"}, {"source": "Image Colorizer", "target": "Image-to-Text"}, {"source": "Image Search (by Image)", "target": "Image Style Transfer"}, {"source": "Image Stitcher", "target": "Image Style Transfer"}, {"source": "Image Style Transfer", "target": "Image Colorizer"}]} |
> | Self-Instruct | "sampled_nodes": [{"id": "Audio-to-Image", "input-type": ["audio"], "output-type": ["image"]}, {"id": "Image Colorizer", "input-type": ["image"], "output-type": ["image"]}, {"id": "Image Stitcher", "input-type": ["image", "image"], "output-type": ["image"]}, {"id": "Image Style Transfer", "input-type": ["image", "image"], "output-type": ["image"]}, {"id": "Image-to-Text", "input-type": ["image"], "output-type": ["text"]}],  | I have a black and white image of an old newspaper (example.jpg) and I want to colorize it first, then apply the style of a modern newspaper (example.png) to it, and finally extract the text from the stylized image. | {"nodes": [{"id": "Image Colorizer", "input": ["example.jpg"]}, {"id": "Image Style Transfer", "input": ["Image Colorizer", "example.png"]}, {"id": "Image-to-Text", "input": ["Image Style Transfer"]}], "links": [{"source": "Image Colorizer", "target": "Image Style Transfer"}, {"source": "Image Style Transfer", "target": "Image-to-Text"}]} |
>
> *Table 2. Comparative analysis of Back-Instruct, Back-Instruct w/o edges, and Self-Instruct.*

---

> ### Author Response · Authors · 2023-11-20
> **General Response - 2**
>
> ### **[G2] Need more error analysis and case studies on dataset construction and predictions**
>
> ### **Response:**
>
> Thank you for the insightful comments. In response, we have conducted a detailed error analysis on both the TaskBench dataset and the predictions made by various LLMs. Our findings are outlined below:
>
> 1. **Error Analysis on TaskBench Dataset**
>
> Despite the advanced instruction generation and labeling capabilities of GPT-4, we admit that it is challenging to guarantee the correctness of all generated samples. To better understand our dataset and assess its accuracy, following the suggestions of Reviewer LVvj and CSeK, we conduct human evaluations to provide a thorough error analysis. Here, we first randomly sampled 148 samples, and our labeling team identified 18 error samples (nearly 12%) from the sampled data. We attribute these incorrect samples to five distinct error categories, and concrete examples of each are presented in Table 3:
>
> - **Incorrect Instructions**
>     - **Incomplete instructions**: This error occurs when the instructions lack the necessary details or resources for successful completion.
>     - **Impractical instructions**: The instructions could be irrational or impossible to execute with the capabilities of current tools.
> - **Parameter Errors**
>     - **Mismatched parameter types**: This error occurs when the parameters provided do not match the expected types for the used tool.
>     - **Incorrect parameter value**: This error is evident when the values provided for the parameters are incorrect or not suitable for the task.
> - **Incorrect Tool Dependency**: This error type refers to the incorrect linking or sequencing of tools required for a task.
>
> | Error Type | Example |
> | --- | --- |
> | Incomplete instructions: missing external resources | Instruction: I have a long text and I would like to get a summarized version of it, then generate an image that represents the main idea of the summarized text. |
> | Impractical instructions: irrational tool invocation patterns | Instruction: I have a text: 'This training vid is amazing! Speed it up by 1.5x please!'. Analyze the sentiment, expand it, find the video URL and adjust the video speed. |
> | Mismatched parameter types | Instruction: I want to find articles related to climate change and analyze their sentiment. Please translate non-English articles to English. Tool invocation graph: {"task_nodes": [{"task": "Text Search", "arguments": ["climate change"]}, {"task": "Text Sentiment Analysis", "arguments": ["<node-0>"]}, {"task": "Text Translator", "arguments": ["<node-1>"]}], "task_links": [{"source": "Text Search", "target": "Text Sentiment Analysis"}, {"source": "Text Sentiment Analysis", "target": "Text Translator"}]} |
> | Incorrect parameter value | Instruction: I have two audio files from online lectures at the following URLs: 'example1.wav' and 'example2.wav'. I want them combined into a single audio file, transcribe the speech into text, and check the text for grammatical errors. Tool invocation graph:  {"task_nodes": [{"task": "Audio Downloader", "arguments": ["example1.wav", "example2.wav"]}, {"task": "Audio Splicer", "arguments": ["<node-0>"]}, {"task": "Audio-to-Text", "arguments": ["<node-1>"]}, {"task": "Text Grammar Checker", "arguments": ["<node-2>"]}], "task_links": [{"source": "Audio Downloader", "target": "Audio Splicer"}, {"source": "Audio Splicer", "target": "Audio-to-Text"}, {"source": "Audio-to-Text", "target": "Text Grammar Checker"}]} |
> | Incorrect Tool Dependency | Instruction: I want to create a more engaging version of this short text: 'Join us for a fun-filled evening!' and find some videos related to its sentiment. Tool invocation graph:  {"task_nodes": [{"task": "Article Spinner", "arguments": ["<node-2>"]}, {"task": "Text Expander", "arguments": ["Join us for a fun-filled evening!"]}, {"task": "Text Sentiment Analysis", "arguments": ["<node-1>"]}, {"task": "Video Search", "arguments": ["<node-2>"]}], "task_links": [{"source": "Text Expander", "target": "Text Sentiment Analysis"}, {"source": "Text Sentiment Analysis", "target": "Article Spinner"}, {"source": "Text Sentiment Analysis", "target": "Video Search"}]} |
>
> *Table 3. Error Analysis on Back-Instruct Dataset.*
>
> Based on these observed errors, we conclude that it is necessary to build a more elaborate prompt (e.g., more detailed tool-use specification and demonstrations) to describe tool parameters and tool dependencies when generating the tool invocation graph. Besides, we will also introduce more high-quality criteria to continuously improve our dataset in addition to our rule-based and LLM-based critics.

---

> > ### Author Response · Authors · 2023-11-20
> > **General Response - 3**
> >
> > 2. **Error Analysis of different LLMs in predicting tool invocation graph**
> >
> > Moreover, we also analyze the failures of different LLMs in tool invocation graph parsing that occur during task automation inference. These failures can be categorized into three main groups: incorrect tool names, incorrect tool dependencies, and mismatched tool parameters. For our analysis, we randomly selected 50 predictions, and the distribution of each error type across different LLMs is detailed in Table 4. We observed that:
> >
> > - gpt-4 demonstrates the fewest errors in all categories, indicating a higher accuracy in tool invocation graph prediction.
> > - gpt-3.5-turbo and code-llama-13b show a progressively higher number of errors. Notably, the 'Tool Parameter Error' is the most common across all models, highlighting the challenge LLMs face in predicting accurate parameters for tools.
> >
> > |  | Required Tool Missing | Tool Dependency Error | Tool Parameter Error |
> > | --- | --- | --- | --- |
> > | gpt-4 | 0 | 2 | 3 |
> > | gpt-3.5-turbo | 2 | 8 | 11 |
> > | code-llama-13b | 4 | 9 | 13 |
> >
> > *Table 4. Error Distribution in Different LLMs*
> >
> > Further, we present specific cases to elucidate the nature of prediction errors in these LLMs. Given the following example, gpt-4 correctly interpreted the task in the given example, underscoring its advanced task automation capabilities. Conversely, gpt-3.5-turbo and code-llama-13b omitted a critical tool ('Audio Downloader'), resulting in a 'Missing Required Tool' error. Additionally, code-llama-13b encountered compounded errors, including 'Tool Parameter Error' and 'Incorrect Tool Dependency'. We have added these cases to our revised version.
> >
> > > Instruction:
> > I need an audio file downloaded from 'https://www.example.com/example.wav', then please reduce the background noise and apply a reverb effect according to my instruction 'reverb 50%'. Finally, combine it with the audio file 'example.wav'.
> > >
> >
> > > Gold Tool Invocation Graph:
> > {"task_nodes": [{"task": "Audio Downloader", "arguments": ["https://www.example.com/example.wav"]}, {"task": "Audio Noise Reduction", "arguments": ["<node-0>"]}, {"task": "Audio Effects", "arguments": ["<node-1>", "reverb 50%"]}, {"task": "Audio Splicer", "arguments": ["<node-2>", "example.wav"]}]}*
> > >
> >
> > | LLM | Predicted Tool Invocation Graph | Error Type |
> > | --- | --- | --- |
> > | gpt-4 | {"task_nodes": [{"task": "Audio Downloader", "arguments": ["https://www.example.com/example.wav"]}, {"task": "Audio Noise Reduction", "arguments": ["<node-0>"]}, {"task": "Audio Effects", "arguments": ["<node-1>", "reverb 50%"]}, {"task": "Audio Splicer", "arguments": ["<node-2>", "example.wav"]}]} | None |
> > | gpt-3.5-turbo | {"task_nodes": [{"task": "Audio Noise Reduction", "arguments": [""https://www.example.com/example.wav""]}, {"task": "Audio Effects", "arguments": ["<node-0>", "reverb 50%"]}, {"task": "Audio Splicer", "arguments": ["<node-1>", "example.wav"]}]} | Missing Required Tool |
> > | code-llama-13b | {"task_nodes": [{"task": "Audio Noise Reduction", "arguments": ["example.wav"]}, {"task": "Audio Effects", "arguments": ["<node-0>", "reverb 50%"]}, {"task": "Audio Splicer", "arguments": ["<node-1>", "<node-0>"]}]} | Missing Required Tool & Tool Parameter Error & Incorrect Tool Dependency: |
> >
> > *Table 4. Case Studies of Prediction Errors in LLMs*

---

> ### Author Response · Authors · 2023-11-20
> **General Response - 4**
>
> ### **[G3] Comparison with ToolBench [1-2]**
>
> ### **Response:**
>
> Following the suggestions of Reviewer BEYM and peHW, we also conducted a detailed analysis to discuss the differences between ToolBench and TaskBench (please see Appendix A.1). We summarize these differences from the perspectives of datasets and evaluations:
>
> 1. **Datasets:** Although both ToolBench and TaskBench have adopted instruction methods to synthesize samples, the mechanisms of these two methods are different. For ToolBench, it just randomly samples some APIs and then uses LLMs to prompt them to generate the final instructions. As a result, it cannot obtain the ground truth and then involve Solution Path Annotation which uses a depth-first search-based decision tree to obtain answers. However, such a strategy also brings new issues like costs and hallucination or bias in answer annotations. However, due to the design of Tool Graph, TaskBench demonstrates more advantages when compared with ToolBench:
>     - **Efficiency**: Our method requires only one API call to generate a complete sample (creating instructions, generating a tool invocation graph, and checking). In contrast, ToolLLM requires one API call to generate instructions and an average of four inference steps during DFS for annotation. Additionally, ToolLLM uses few-shot learning, which consumes more tokens than our zero-shot approach.
>     - **Higher Quality of Instructions**: Our tool graph includes potential tool invocation relationships (resource and temporal dependencies). Based on the tool graph, the instructions generated by ChatGPT are more in line with actual human behavior. As we demonstrated in our [G1] human evaluation, the generation based on a tool graph with edges significantly improves Naturalness, Complexity, and Alignment compared to generation without edges.
>     - **Stability**: ToolLLM’s instruction generation might not cover all sampled instructions based on the API set. Our method not only covers all sampled tools but also strictly follows the dependency between tools. We can control instruction generation through the sampling process, such as the number of tools and the topology of the tool graph.
>     - **Reliable Annotations**: In ToolLLM, instruction generation, and tool invocation annotation are independent processes.  However, In our approach, the sampled tool graph can directly be utilized as the annotations for the final assessment. Hence, our Back-Instruct can directly generate instructions with better alignments to the annotations and does not need to generate answers anymore. Besides, we also utilize the consistency between the sampled tool graph and the generated tool invocation graph to further filter out low-quality annotations, ensuring high-quality tool graph annotations.
> 2. **Evaluation:** ToolBench introduces ToolEval in their paper. However, ToolEval just directly follows AlpacaEval, which uses win rate and pass rate to evaluate the capability of LLMs, and cannot manifest any characteristics of LLMs in tool utilization and planning. Our evaluation encompasses multiple aspects, including task decomposition, tool prediction, and parameter prediction. For each aspect, we have developed multiple well-designed metrics to assess the different capabilities required for task automation.
>
> [1] Tool learning with foundation models. Qin et al. 2023.
>
> [2] ToolLLM: Facilitating Large Language Models to Master 16000+ Real-world APIs. Qin et al. 2023.

---

### Author Response · Authors · 2023-11-20
**To All Reviewers**

We sincerely thank each reviewer for providing their constructive comments, which are very helpful in improving our paper. Below are our modifications to the latest version:

1. Following suggestions from Reviewers LVvj, peHw, and CSek, we have added human evaluations of the TaskBench dataset in Appendix A.3.
2. Following suggestions from Reviewers LVvj, BEYM, and  CSek, we included a case study and error analysis of the TaskBench dataset construction in Appendix A.4.1.
3. Following suggestions from Reviewers LVvj and Reviewer BEYM, we have added a case study and error analysis of LLMs in task automation in Appendix A.4.2.
4. We have added Appendix A.2 to provide a detailed comparison of our benchmark with ToolBench.
5. Following the suggestion from Reviewer BEYM, we have expanded the experimental analysis in Section 3.5.
6. The analysis in the few-shot setting has been moved to Appendix A.6.2.
7. We have added a visualization of the tool graph in A.10.
8. We have included experimental results in Appendix A.8.1 to demonstrate the relationship between task automation performance and the number of tools.
9. We have added the task automation performance of Claude-2 in Appendix A.7.

---

### Author Response · Authors · 2023-11-22
**To All Reviewers**

Thanks for all your constructive comments, which have significantly improved the quality of our datasets. We have completed the verification of our datasets, and updated them to the latest version (please see supplemental material).

Following the suggestions of each reviewer, we have conducted the following strategies to improve our datasets:

1. We use LLM-based and Rule-based Critics to ensure the validity of the generated tool graphs.
2. We invited 12 human annotators to evaluate the quality of each sample in our dataset, considering aspects such as grammatical correctness, completeness of instructions, executability of the tool invocation graphs, and their alignment. Based on our examinations, we will first discard cases that are severely incorrect or do not fit real-world scenarios. Then, for cases with minor errors (e.g., syntax), we will ask the annotators to fix them.

We report the statistics of the dataset processing in the following tables:

| Domain | #Samples | #Samples Checked by Critics (%) | #Samples Verified by Humans (%) |
| --- | --- | --- | --- |
| HuggingFace Tools | 12,217 | 8,457 (69.22%) | 8,104 (66.33%) |
| Multimedia Tools | 8,904 | 6,281 (70.54%) | 5,887 (66.13%) |
| Dailylife APIs | 7,150 | 5,432 (75.97%) | 4,350 (60.84%) |

| Domain | #Samples | #Checked by LLM-based Critics (%) | #Checked by Rule-based Critics (%) | #Checked by Both Critics (%) |
| --- | --- | --- | --- | --- |
| Hugging Face Tools | 12,217 | 9,042 (74.01%) | 10,289 (84.22%) | 8,457 (69.22%) |
| Multimedia Tools | 8,904 | 6,959 (78.16%) | 7,363 (82.69%) | 6,281 (70.54%) |
| Dailylife APIs | 7,150 | 5,694 (79.63%) | 6,271 (87.70%) | 5,432 (75.97%) |

| Domain | #Samples Checked by Critics | #Correct Samples (%) | #Discarded (%) | #Fixed for Syntax (%) | #Fixed for Instructions (%) | #Fixed for Tool Invocation Graph (%) |
| --- | --- | --- | --- | --- | --- | --- |
| Hugging Face Tools | 8,457 | 6,974 (82.46%) | 353 (4.17%) | 27 (0.32%) | 328 (3.87%) | 843 (9.96%) |
| Multimedia Tools | 6,281 | 5,262 (83.77%) | 394 (6.27%) | 11 (0.17%) | 107 (1.70%) | 526 (9.96%) |
| Dailylife APIs | 5,432 | 4,307 (79.29%) | 684 (12.59%) | 6 (0.11%) | 92 (1.68%) | 332 (6.11%) |

In the supplemental material, our TaskBench contains datasets in three areas: HuggingFace Tools, Multimedia Tools, and Dailylife APIs. Each dataset directory includes three types of files:

- `data_formulated.json` is the raw dataset generated by GPT-4. The file we submitted in the previous version.
- `data_critics.json` is the dataset obtained after checking and filtering by rule-based and LLM-based critics. Merged from the `data_formulated.json` and `alignment_ids.json` files, which were submitted in the previous version.
- `data_human.json` is the latest human-verified version. We invited a dozen human annotators to closely check and fix the samples to ensure the quality of the dataset.

For more details, please refer to the folder `datasets_human_verified` in our supplementary material.

---

### Author Response · Authors · 2023-11-23
**To All Reviewers**

We sincerely thank each reviewer for their continuous constructive comments, which have helped us improve the quality of TaskBench. Compared to previous works, TaskBench not only provides self-instruct samples, but also fine-grained annotations (e.g., decomposed sub-tasks, tool invocation and their parameters) at each stage of task automation. Therefore, we admit that it requires us to provide more elaborate examinations to polish the final dataset. To fulfill this point and follow the suggestions of each reviewer, we have designed multiple strategies, including LLM-based and Rule-based critic, as well as human verifications to iteratively improve our dataset:

1. LLM-based and Rule-based Critics:

    - Evaluating the Executability of the tool invocation graph using LLMs.
    - Evaluating the alignment of the tool invocation graph using LLMs.
    - Ensuring node set consistency through rule-based checks.
    - Verifying edge set consistency via rule-based methods.
    - Checking for consistency in the number of tool parameters*.
    - Maintaining consistency in tool parameter names*.

2. Human Verifications:
    - Checking grammatical correctness.
    - Ensuring completeness of instructions.
    - Verifying executability of the tool invocation graphs.
    - Assessing alignment.

(* denote the latest update based on the feedback of Reviewer CSek).

Based on the above strategies, we conduct multi-turns verifications to iteratively refine our datasets including:

- Discarding cases that are severely incorrect or do not fit real-world scenarios.
- Fixing Syntax.
- Revising Instructions.
- Correcting the Tool Invocation Graph.
- Adjusting parameters of tools.

We also thank Reviewer CSek for providing continuous feedback and examinations to polish our dataset. We will continue our research to develop more powerful and high-quality datasets in the future.

---

### Meta-Review · Area_Chair_S1fe · 2023-12-05

**Metareview:**

This paper introduces TaskBench, a new benchmark to evaluate task automation capabilities of large language models. Solving TaskBench problems requires an LLM to perform three separate stages: task decomposition (decomposing complex, composition tasks into individual steps), tool invocation (calling the right tool API to solve each step), and parameter invocation (parameterizing tool invocations). Execution of a multi-step task can be represented as a tool graph, with nodes representing tools and edges their dataflow dependency. To construct TaskBench, tool graphs are synthetically generated, and then back-translated to synthetic natural language instructions (so called “back-instruct” in the paper). TaskBench comes with synthetic problems in three domains (HuggingFace model APIs, Multimedia data processing, and daily life services such as web search and shopping, with **103 tools** in total). Those synthetic problems are filtered by self-consistency as well as a series of LLM and rule-based filters. The authors evaluated upto 16 LLMs on TaskBench, and the results suggest that task automation capabilities could be attributed to the general reasoning and instruction following skills of those LLMs.

**Strengths**

* Broad Coverage of Tools: TaskBench comes with 103 tools in total (by the total number of node types on task graphs), with much broader coverage of use cases than existing works (LVvj). Given the burgeoning of LLM tool-use research, TaskBench could become an important benchmark to evaluate task automation skills of LLMs (BEYM, peHw, CSek).

* Task Representation Formalisms: Using tool invocation graphs to capture the dependency and dataflow of tools can represent compositional task execution logics involving multiple tools with complex dependencies (peHw). The back-translation method to generate natural language instructions from tool graphs also improves data collection efficiency (peHw, CSek).

* Comprehensive Evaluations: up to 16 LLMs (CSek).

**Weaknesses**

All reviewers concurred that a major issue of this submission is the dataset quality. Since problems in TaskBench are synthetically generated, some of those tasks can be unnatural and do not align with real user intents (LVvj, peHw, CSek). In addition, there is no guarantee that the back-translated natural language instructions are aligned with the original task graphs (LVvj, peHw, CSek). In particular, upon manual inspection, reviewer CSek spotted several obvious errors in the dataset. While the authors devised a series of model and rule-based filters to reject problematic examples, the coverage and accuracy of those filters remain unclear after the author’s response. Besides, while the authors conducted manual analysis of dataset quality on a subset of TaskBench problems upon request of the reviewers, this subset is too tiny (50-148, Appendix A.3-4) compared to the size of the entire dataset (~20K, Table 1).

Given that this is a benchmark paper and data quality still remains an issue, the decision is a “Reject”. To improve the paper, the reviewers strongly suggest conducting manual review over *all* the examples in the benchmark dataset.

**Justification For Why Not Higher Score:**

Given that this is a benchmark paper and data quality still remains an issue, it would be risky to accept this submission.

**Justification For Why Not Lower Score:**

N/A

---

### Decision · Program_Chairs · 2024-01-16

Reject